# T-Plastin reinforces membrane protrusions to bridge matrix gaps during cell migration

Damien Garbett [1]✉, Anjali Bisaria [1], Changsong Yang[2], Dannielle G. McCarthy [3], Arnold Hayer [1,4], W. E. Moerner [3], Tatyana M. Svitkina [2] & Tobias Meyer [1,5]✉

Migrating cells move across diverse assemblies of extracellular matrix (ECM) that can be separated by micron-scale gaps. For membranes to protrude and reattach across a gap, actin filaments, which are relatively weak as single filaments, must polymerize outward from adhesion sites to push membranes towards distant sites of new adhesion. Here, using micropatterned ECMs, we identify T-Plastin, one of the most ancient actin bundling proteins, as an actin stabilizer that promotes membrane protrusions and enables bridging of ECM gaps. We show that T-Plastin widens and lengthens protrusions and is specifically enriched in active protrusions where F-actin is devoid of non-muscle myosin II activity. Together, our study uncovers critical roles of the actin bundler T-Plastin to promote protrusions and migration when adhesion is spatially-gapped.

[1] Department of Chemical and Systems Biology, Stanford University, Stanford, CA, USA. [2] Department of Biology, University of Pennsylvania, Philadelphia, PA, USA. [3] Department of Chemistry, Stanford University, Stanford, CA, USA. [4] Present address: Department of Biology, McGill University, Montréal, Canada. [5] Present address: Department of Cell and Developmental Biology, Weill Cornell Medicine, New York, NY, USA. ✉email: dgarbett@stanford.edu; tom4003@med.cornell.edu

Tissues and organs contain different orientations and spacing of extracellular matrix (ECM), which in turn directs critical cellular behaviors such as migratory path, proliferation, and the axis of cell division[1,2]. The ECM is often non-uniform, containing micron scale gaps, which cells must migrate across during development, angiogenesis, and tumor metastasis[2–4]. The vascularization of endothelial cells during developmental and pathogenic angiogenesis are dependent on the distribution, formation, and breakdown of ECMs (fibronectin, collagen, laminin, fibrin, and others) in the retina[5], and many other tissues[6]. Nevertheless, most in vitro cell migration assays have utilized uniform ECM distributions or gradients of ECM concentration. Recent advancements in light-induced protein adsorption allow for the generation of ECM patterns with sub-micron resolution[7], and facilitates investigations of how cells can bridge adhesion gaps during cell migration.

The actin cytoskeleton is arguably the most important structure supporting membrane protrusions and cell migration. Large linear contractile filaments, called stress fibers, are bundled in part by myosin motor proteins that allow cells to strengthen integrin-based adhesions and also to retract their rears as they move forward. In protrusions at the front of cells, such as lamellipodia and filopodia, actin polymerization forces help push the plasma membrane forward. These protrusive forces are opposed by counteracting membrane tension[8,9].

Cells utilize integrin-based focal adhesions to connect the actin network to the underlying ECM. First small nascent adhesions form along the edge of protrusions, such as lamellipodia and filopodia. Over time, a subset of these nascent adhesions mature into larger focal adhesions further back in the lamellum and ultimately end up in the rear of cells during migration. Mature adhesions are strengthened by stress fibers and myosin II-dependent tension[10]. The main roles of focal adhesions are to couple forces generated by the actin cytoskeleton to the external environment and enable cells to protrude membranes in the front and retract membranes in the back during cell migration.

Simply pushing the membrane forward might seem to be a trivial task[11]. However, unsupported actin filaments alone can be bent and lack the strength to polymerize and push against membrane tension, and therefore require other factors for cells to protrude[12,13]. In filopodial protrusions (thin finger-like actin-based protrusions), fascin is a well-characterized actin bundler[14]. However, fascin is dispensable for both developmental and tumor angiogenesis[15], and does not bundle the larger branched actin network in lamellipodia[16]. Arp2/3-based actin filament nucleation activity creates the broader, branched actin network with a high number of distal ends and typically drives membrane protrusions during cell migration. This branched network requires nearby nascent focal adhesion anchors to convert actin polymerization at the barbed ends into a pushing force against opposing membrane tension[17]. Biophysical reconstitution experiments have shown that actin-bundling proteins can increase the stiffness of actin networks in vitro[18]. We hypothesized that when cells encounter ECM gaps, an unknown actin-bundling process may be needed to stabilize single actin filaments in order to strengthen the actin network during membrane protrusion to allow cells to bridge these gaps. However, whether peripheral actin-bundling proteins are important in broad forward protrusions during cell spreading and cell migration was not known.

Here, we generate micropatterns of fibronectin with variable non-adhesive gaps at micron-level scales to examine how cells can protrude across gaps in ECM. We identify the actin-bundling protein T-Plastin as having a critical role in promoting membrane protrusions and enabling cells to bridge ECM gaps to facilitate cell spreading and cell migration.

## Results

**Cells utilize lamellipodia and filopodia to protrude across ECM gaps.** Our study was motivated by super-resolution microscopy of human umbilical cord vascular endothelial cells (HUVEC) that were stained for F-actin and paxillin to visualize actin filaments and focal adhesion contacts within lamellipodia (Fig. 1a). Nascent focal adhesions decorated the entire leading edge of lamellipodia and larger mature focal adhesions were located ~2–5 μm further back in the lamellum at the ends of stress fibers. Since nascent focal adhesions have been linked to protrusion[19], we hypothesized that these outermost nascent adhesions provide local structural rigidity by coupling the protrusive actin network to the underlying ECM to counter the resisting force of membrane tension during protrusion (Fig. 1b). Such an architecture with adhesion support close to the leading edge may allow cells to effectively migrate on uniform ECM.

However, the proximity of nascent adhesions to the leading membrane edge of cells on uniform ECM raises two interesting questions: (i) can a cell still protrude if it encounters an ECM gap and (ii) is there a mechanism that strengthens the actin network in the front to facilitate bridging such gaps? To investigate if and how cells protrude membranes across ECM gaps, we generated micropatterned stripes of fibronectin 2 μm in thickness separated by gaps of PLL-PEG, to which cells cannot adhere, ranging from 4-16 μm in size. The pattern design is similar to ladders with each adhesive fibronectin stripe representing a rung (Figs. 1c and S1a).

HUVEC stably expressing F-tractin-mCitrine, an F-actin reporter[20], were added as a cell suspension to micropatterns of fluorescently-labeled fibronectin and imaged every 30 s for 4 h (Fig. 1d). Markedly, single cells spread and formed lamellipodia that reached across 4 μm gaps in a process that seemed indistinguishable from those typically observed in cells plated on continuous ECM (Figs. 1e, f left and S1b, S1d and movie S1). Interestingly, to cross larger gaps of 6–8 μm, cells often extended filopodia and made contact with a neighboring stripe of fibronectin before a wave of F-actin polymerization expanded the membrane forward as a broad protrusion (Figs. 1e, f middle and S1d and movie S2). On larger gaps of 10–16 μm, cells often failed to cross the gaps and started to exhibit dynamic membrane blebs (Figs. 1e, f right and S1d and movie S3). As a control, cells that settled in areas devoid of fibronectin failed to spread or adhere over long periods of time (Fig. S1c). Quantification of cell spreading areas across these ladder micropatterns showed that as the gap width between ECM stripes increased, cells have increasing difficulty to spread (Fig. 1g).

To determine if protrusions across ECM gaps differ from those on continuous ECM, we examined the formation of local adhesions on these ladder patterns by fixing cells after imaging and then staining for F-actin and paxillin. Cells on these ladder patterns formed stress fibers across gaps of ECM and adhesions on fibronectin stripes that appear similar to what is seen on continuous ECM and other in vitro migration and micropattern assays (Fig. 1h)[21]. Furthermore, since Cdc42 is an important regulator of cell polarity during cell migration[22], we also expressed a FRET reporter for Cdc42 activity[23] and found that Cdc42 was active in protrusions across gaps in ECM and lower in non-protrusive areas of the cell, indicating that Cdc42 signaling remains polarized similar to cells migrating on uniform ECM (Fig. 1i). Together, these experiments suggest that protrusion across ECM gaps utilizes a similar signaling and adhesion machinery to that of cells on continuous ECM.

Since cells were still able to protrude across ~4–8 μm wide ECM gaps, sizes larger than the typical distance between nascent and mature adhesions in lamellipodia (Fig. 1a)[24], we

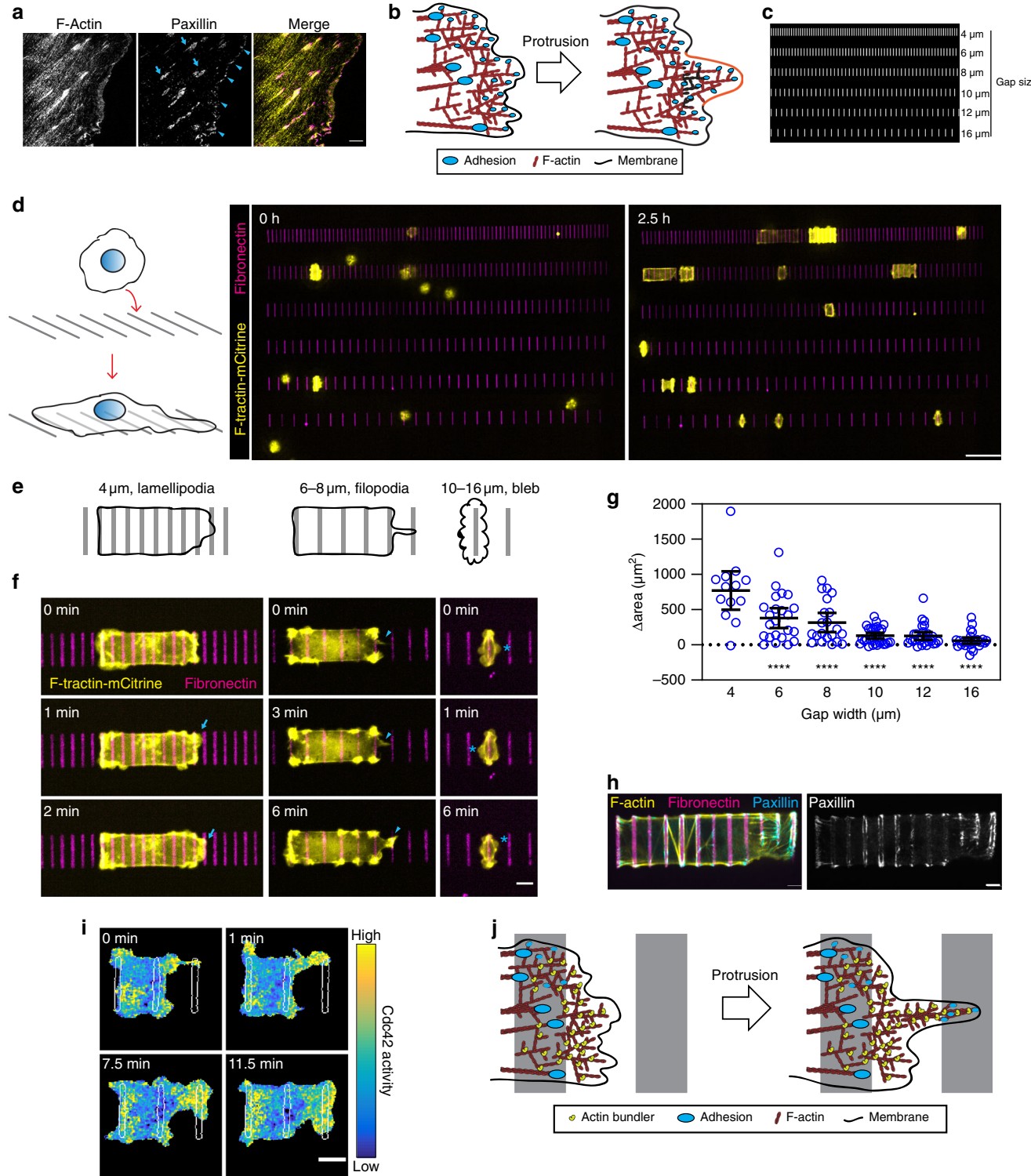

hypothesized that the actin network in the gap region where adhesions cannot form may need to be strengthened since individual filaments can be readily bent[12]. As a working model, we considered that actin-bundling proteins may have a key role in strengthening the protrusive actin network to promote cell spreading and cell migration when cells encounter gaps in ECM (Fig. 1j). Furthermore, since membrane tension increases as cells spread over a larger area, we further considered that such a strengthening of the actin network also allows cells to protrude membranes at higher relative membrane tension.

**T-Plastin is an ancient actin bundler required for effective lamellipodial protrusion**. To determine whether actin-bundling protein(s) promote membrane protrusion across gaps in ECM we first examined the localizations of 7 candidates that were selected based on available RNA sequencing datasets of endothelial cells (Fig. S2a)[25]. Out of these endothelial actin-bundling proteins, only T-Plastin was selectively localized to lamellipodia in HUVEC (Figs. 2a and S2b). Moreover, T-Plastin was also enriched in lamellipodia of mouse melanoma cells (B16-F1) and human epithelial cells (RPE-1) (Fig. S2c).

**Fig. 1 Use of a ladder-type ECM micropattern to investigate how cells can protrude across gaps in ECM. a** 3D-SIM of HUVEC lamellipodia, F-actin (yellow) and paxillin (magenta). Mature and nascent focal adhesions highlighted with cyan arrows and arrowheads, respectively, bar 5 μm. **b** Model of protrusion; nascent adhesions stabilize F-actin at the leading edge against opposing forces from membrane tension during a protrusion (orange membrane). **c** Example of ladder patterns of fibronectin (white) with gaps ranging from 4 to 16 μm coated with non-adhesive PLL-PEG (black). **d** Left, schematic: cells added to ladder patterns and protrude across gaps between fibronectin stripes as they spread. Right, HUVEC expressing Ftractin-mCitrine (yellow) added to fibronectin patterns (magenta). Bar, 50 μm. **e** Examples of cellular protrusions across different gap sizes. Smaller gaps of 4 μm, cells protrude broad lamellipodia. Larger gaps of 8 μm, cells utilize filopodia to initially protrude and then expand after making contact with the next fibronectin rung. Very large gaps of 12 μm cells, bleb and struggle to cross gaps or reach neighboring fibronectin rungs. **f** Images showing cellular behaviors depicted in (**e**). Lamellipodia, filopodia, and blebs are highlighted with cyan arrows, arrowheads, and asterisks, respectively, bar 10 μm. **g** The change in cell surface area one hour after first making contact with a stripe of fibronectin was quantified. Black bars represent mean and 95% confidence intervals (n of 13, 23, 21, 31, 28, 26 cells for gaps of 4, 6, 8, 10, 12, and 16 μm, respectively from greater than three independent replicates), each gap size was compared to 4 μm using a one-way ANOVA with Dunnett's multiple comparison test ****$P < 0.0001$. **h** Cells on patterns were fixed and stained for F-actin (yellow), paxillin (cyan); bar 5 μm. **i** Cells expressing Raichu Cdc42 FRET sensor (parula colormap, yellow and blue are 99th (high), 3rd (low) percentiles) were imaged on fibronectin patterns (white), bar 10 μm. **j** Model of protrusive actin-bundling playing an important role to compensate for lack of adhesion when cells protrude across non-adhesive gaps in order to reach the next available region of ECM.

T-Plastin, along with α-actinin and myosin, represent the oldest known actin-bundling proteins[26,27] and actin-bundling is the only established cellular activity of plastins[28]. There are three Plastin isoforms with distinct tissue distributions in mammals with homologs found in distant eukaryotes such as plants and fungi (Fig. S2d). Plastin actin-bundling activity has been extensively characterized in vitro and is regulated by their $Ca^{2+}$-binding EF-hand domains[26,29]. However, T-Plastin is less sensitive to $Ca^{2+}$ inhibition than its closest family member L-Plastin[30] (Fig. S2d). In HUVEC that collectively migrate into open space, endogenous T-Plastin localized specifically to lamellipodia that reach into cell-free areas as well as to cryptic lamellipodia that reach under neighboring cells during cell migration (Fig. 2b). Knockdown of T-Plastin with siRNA reduced the amount of F-actin staining in lamellipodia and also resulted in more jagged leading edges compared to HUVEC treated with control siRNA (Fig. 2c), consistent with T-Plastin having a role in reinforcing the actin network in membrane protrusions. Arp2/3 is an actin nucleator important for generating branched actin networks in lamellipodia[31], and its enrichment at the leading edge was also greatly reduced in T-Plastin knockdown cells (Fig. 2d), indicating that it is the protrusive actin network that is selectively diminished in the absence of T-Plastin.

We next examined the contractile actin network, which can be visualized in cells expressing fluorescently-tagged myosin light chain (MYL9), and confirmed that it is nearly completely depleted from the front of migrating cells[32]. Cells expressing both T-Plastin-mCitrine and MYL9-mTurquoise showed a striking spatial separation of the respective proteins, with T-Plastin selectively enriched in lamellipodia, and MYL9 localized further back with stress fibers (Fig. 2e). When we knocked down T-Plastin in cells expressing both MYL9-mTurquoise and F-tractin-mCitrine, we observed a marked reduction in the size of the F-actin zone depleted of myosin at the front (Fig. 2f, g). Taken together, these results show that T-Plastin selectively localizes to and strengthens the protrusive actin network at the front of cells.

**T-Plastin enhances the rate of protrusion of the leading edge**. To gain insights how T-Plastin facilitates protrusions, we examined leading edge dynamics in sparsely plated HUVEC expressing F-tractin-mCitrine and treated with siRNA targeting T-Plastin. Cells treated with non-targeting control siRNA showed broad protrusions with dynamic protrusion retraction cycles (Fig. 3a). Cells treated with two different siRNAs targeting T-Plastin showed overall less protrusive regions that were often smaller compared to those seen in control cells. To quantify leading edge dynamics, we generated kymographs and measured the rates of protrusion and retraction cycles (Fig. 3b)[33]. Control cells

protruded with an average rate of ~1.5 μm/min compared to ~1 μm/min in cells treated with either siRNA against T-Plastin (Fig. 3c). Retraction rates of protrusions in si-T-Plastin treated cells were also slower than that in control cells, averaging ~ -1.2 μm/min and -2 μm/min, respectively (Fig. 3d). We conclude that T-Plastin promotes membrane protrusions in part by enhancing the rate of at which protrusions are extended. This is consistent with the interpretation that T-Plastin has an actin-bundling role that strengthens the actin network and allows more force to be projected to push the membrane outward against an opposing membrane tension.

**T-Plastin is selectively enriched in actively protruding lamellipodia**. We next examined the localization of T-Plastin in live cells expressing T-Plastin-mRuby3 and F-tractin-mCitrine to better understand its role in facilitating protrusions. During migration, sub-regions of a lamellipodium can protrude and retract despite an overall net protrusion of the cell front. T-Plastin was highly enriched (shown as the ratio of T-Plastin over F-tractin) in protrusive sub-regions compared to other sub-regions undergoing transient retractions in HUVEC (Fig. 4a and movie S4). We also performed a kymographic analysis of the cell periphery by separating it into ~55 windows and to compare the protrusion/retraction velocity to the level of T-Plastin in each window (Fig. 4b). Markedly, this analysis shows a close correlation between T-Plastin enrichment and protrusive activity in sub-regions of the cell front (Fig. 4c, d). Temporal cross-correlation analysis of T-Plastin and F-tractin intensities with protrusions showed a parallel increase in T-Plastin and F-tractin levels just prior to a protrusion event and persisted for ~15 s after a protrusion (Fig. 4e).

To better understand the relationship between T-Plastin and Arp2/3-mediated actin polymerization, we examined how T-Plastin enrichment is altered when we halt protrusions by using the Arp2/3 inhibitor CK666 in cells expressing T-Plastin-mRuby3, MYL9-mTurquoise, and Ftractin-mCitrine. Addition of 200 μM CK666 not only halted protrusions but also caused an immediate reduction in T-Plastin enrichment at the front (Fig. 4f). As the cell edge retracted, small retraction fibers began to form that were enriched in T-Plastin, while myosin activity became increasingly localized to an area depleted of T-Plastin just behind these retraction fibers. There was a remarkable exclusivity between F-actin labeled by T-Plastin and decorated by myosin, suggesting that the two bundling activities are not compatible with each other within the same F-actin network.

To further examine the dynamic localization of T-Plastin and myosin, we next treated cells with the F-actin stabilizing drug jasplakinolide before adding the Arp2/3 inhibitor CK666 (Fig. 4g).

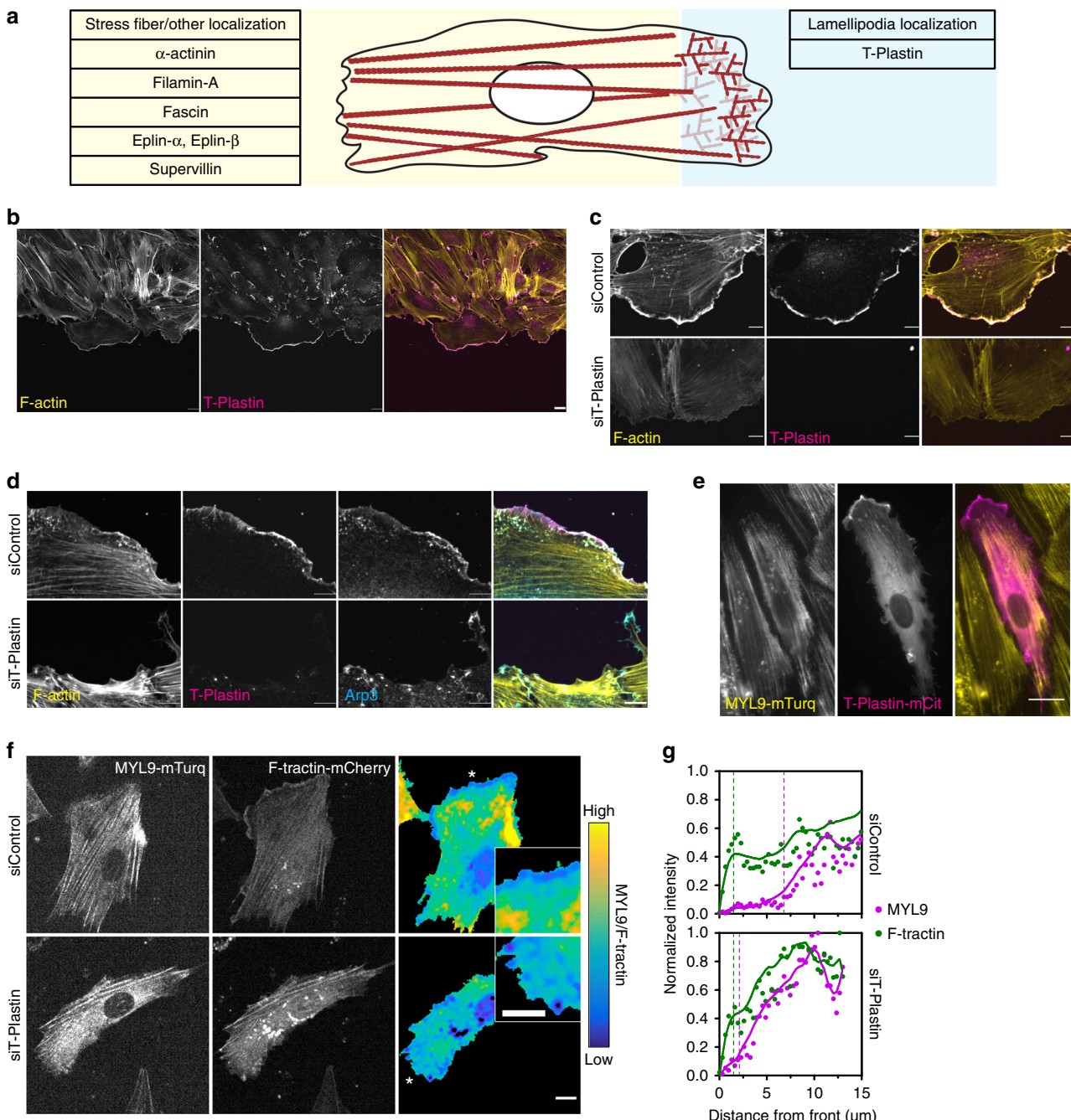

**Fig. 2 Mutual exclusivity of T-Plastin and myosin localizations: T-plastin localizes to and regulates the size of lamellipodia. a** Localization summary of actin-bundling proteins expressed in endothelial cells (localizations shown in Fig. S2a–c). **b** Sheets of HUVEC migrating into a scratch stained for F-actin (yellow) and endogenous T-Plastin (magenta), bar 20 μm. **c** Cells treated with control or T-Plastin siRNA were fixed and stained for F-actin (yellow) and endogenous T-Plastin (magenta), bar 10 μm. **d** Cells treated with control or T-Plastin siRNA were fixed and stained for F-actin (yellow) and endogenous T-Plastin (magenta), and Arp3 (cyan), bar 10 μm. **e** Cells stably expressing MYL9-mTurquoise (yellow) were transfected with T-Plastin-mCitrine (magenta) and imaged, bar 20 μm. **f** Cells expressing F-tractin-mCitrine and MYL9-mTurquoise were treated with control or T-Plastin siRNA and imaged. The ratio of MYL9 over F-tractin is shown as a parula colormap (yellow and blue are 99th (high), 3rd (low) percentiles), bar 10 μm. **g** Quantification of intensities of MYL9 (magenta) and F-tractin (green) along lines drawn from the leading edge into the lamellum of (*n* cells for siControl = 9, siT-Plastin = 10, from two independent replicates) as shown in (**f**). Dots represent the median value for each position, lines are LOWESS spline fits of the data, and dotted vertical lines indicate regions of actin at leading edge and where myosin activity begins.

When F-actin is stabilized and protrusions are halted, T-Plastin remains enriched at the front and myosin activity is simultaneously prevented from accumulating near the cell edge, suggesting that the loss of T-Plastin localization and increased local myosin activation in a protrusion requires a loss of actin stability. We also treated cells with 10 μM of MLCK (myosin light-chain kinase) inhibitor ML-7 (Fig. S3a), which caused an expected loss of myosin activity as seen by the loss of MYL9 localization from stress fibers. ML-7 treatment also caused an increased localization of T-Plastin to newly formed F-actin puncta deficient in myosin activity (Fig. S3a). However, T-Plastin and myosin both remained absent from stress fibers, possibly due

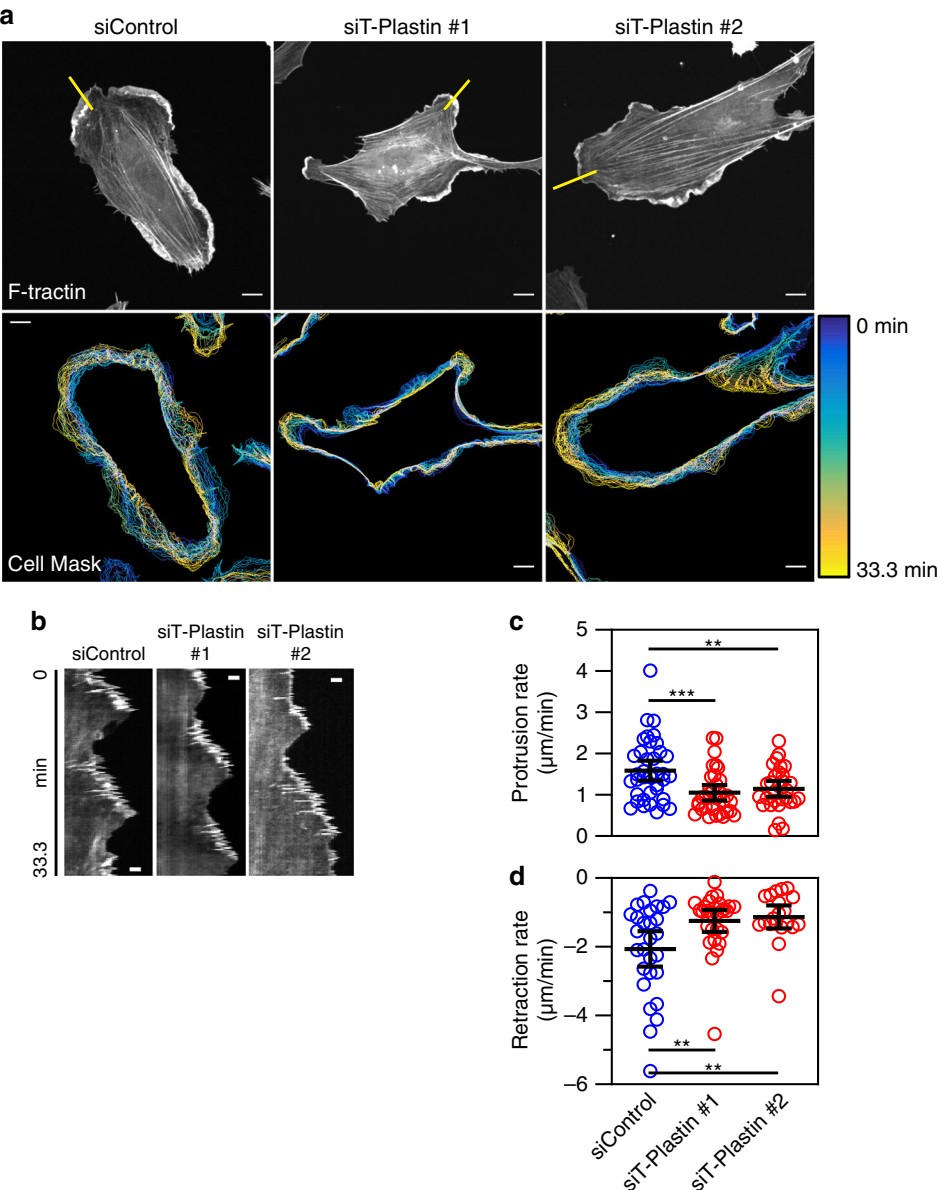

**Fig. 3 T-Plastin enhances protrusion dynamics. a** HUVEC expressing F-tractin-mCitrine were treated with control or T-Plastin siRNA, sparsely plated, and imaged every 10 s for ~33 min. Masks were generated of the cell edge boundaries and time-projected with each line representing a 50 s interval, the parula colormap represents the 0 min time as blue and 33.3 min time as yellow. Straight yellow lines over F-tractin images indicate regions used to generate kymographs shown in (**b**). Bars, 10 μm. **b** Kymographs generated from F-tractin images shown in (**a**), bars 2.5 μm in the x-direction. The y-direction is time with each pixel as 10 s, the entire length equaling 33.3 min. Protrusions can be seen moving outward from the cell body (toward the right) and retractions inward toward the left. **c** Quantification of protrusion rates generated from kymographs such as those shown in (**b**). (n of protrusions are: siControl = 38, siT-Plastin #1 = 34, siT-Plastin #2 = 30, taken from two independent replicates). **d** Quantification of retraction rates generated from kymographs such as those shown in (**b**). (n of retractions are: siControl = 28, siT-Plastin #1 = 28, siT-Plastin #2 = 20, taken from two independent replicates). Bars represent means, error bars represent 95% CI, each siRNA targeting T-Plastin was compared to siControl using a one-way ANOVA with Dunnett's multiple comparison test **$P < 0.01$, ***$P < 0.001$.

to the presence of other bundling proteins such as α-actinin, which has been shown to exclude smaller bundling proteins like fimbrin[34], a member of the plastin family. Finally, treatment of cells with 10 μM of blebbistatin, a broad inhibitor of myosin II, left T-Plastin decorated protrusions initially unaffected but then caused a broadening of lamellipodia after a several minute delay (Fig. S3b). Altogether, these results suggest that (i) T-Plastin has a critical role in strengthening the Arp2/3 generated actin network to promote protrusion, and (ii) that non-muscle myosin II and T-Plastin spatially exclude each other with actomyosin contractility restricting T-Plastin-mediated protrusion.

**Cells can bridge ECM gaps only if T-Plastin activity and focal adhesion strength suffice to counter the opposing force of membrane tension.** We next added HUVEC expressing T-Plastin-mRuby3 and F-tractin-mCitrine onto fluorescently-labeled fibronectin ladders to examine dynamic changes in T-Plastin localization during ECM gap bridging events. The goal of these experiments was to better understand if and how T-Plastin promotes membrane protrusion when cells must cross gaps in ECM. We found that T-Plastin is highly enriched in both filopodia and lamellipodia that protrude across gaps between the fibronectin stripes (Fig. 5a and movie S5). Endogenous T-Plastin also localized

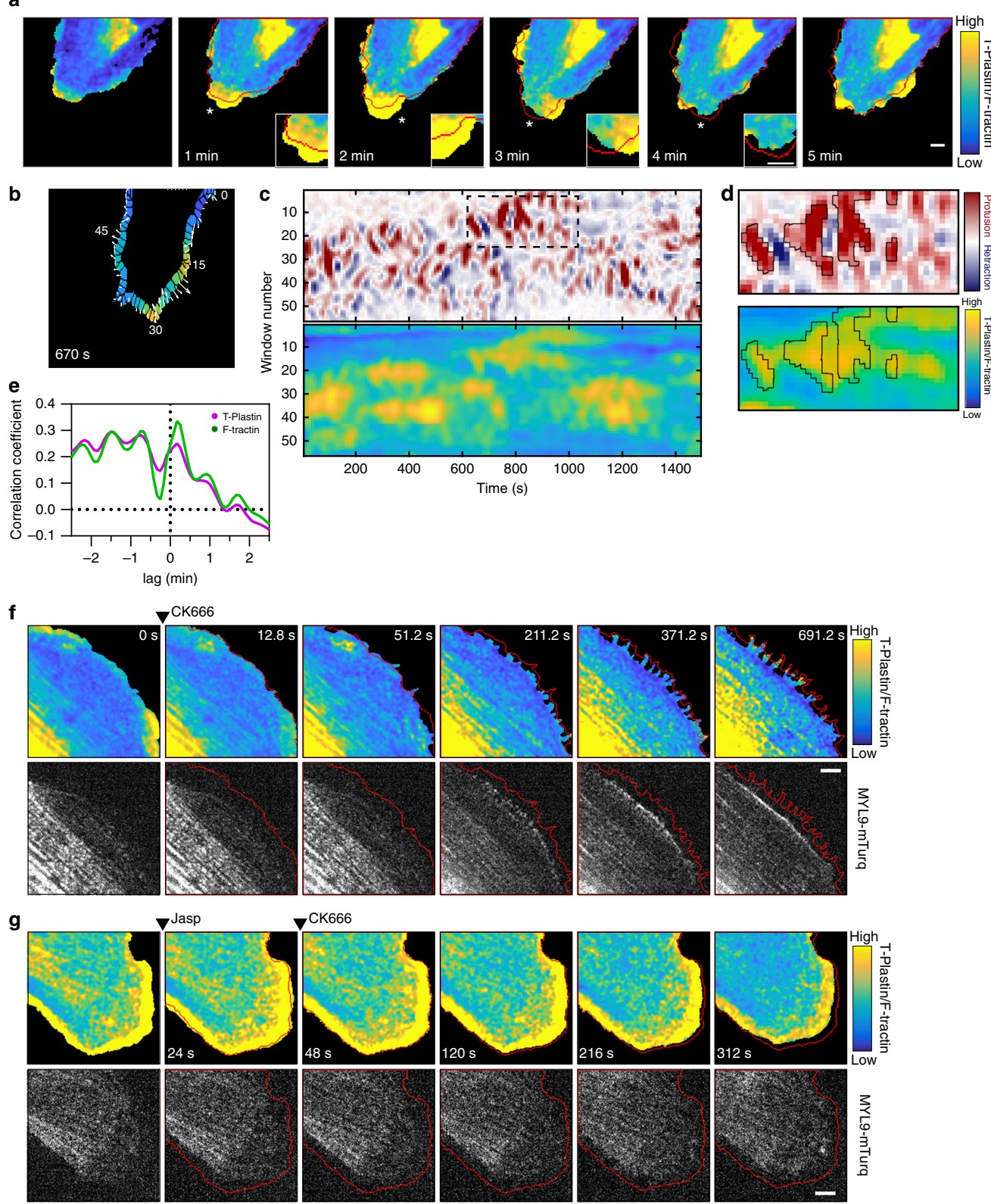

to protrusions in fixed HUVEC on the fibronectin ladder patterns (Fig. S4a, b).

To determine if T-Plastin is important for ECM gap bridging, we compared cell spreading abilities of the three stable CRISPR/Cas9 T-Plastin knockout (KO1-3) HUVEC clones (Fig. S4c, d) to wild-type cells (WT) on ECM ladder micropatterns. As a control, we first ensured that KO and WT cells were similar in size using

cytometry measurements of cell diameter (Fig. S4e). T-Plastin KO cells formed lamellipodial and filopodial protrusions (Figs. S4f and 1f, movies S1, S2, S6, and S7) and the observed focal adhesions appeared to be similar to WT cells (Figs. 5b and S4g) when spreading across non-adhesive gaps in ECM of 4 or 8 μm. However, a quantitative analysis of the degree of cell spreading on the 4 and 8 μm gapped fibronectin ladder patterns showed that

**Fig. 4 Selective association of T-Plastin with extending but not retracting protrusions. a** HUVEC expressing T-Plastin-mRuby3 and F-tractin-mCitrine were plated sparsely and imaged over time. The ratio of T-Plastin over F-tractin is shown as a parula colormap with yellow and blue indicating high (99th percentile) or low (3rd percentile) enrichment of T-Plastin, respectively and is the same scale in all panels using the parula colormap. The cell edge from each previous timepoint image is shown as a red outline; protrusions are present outside the red line while retractions lie within. Subregions blown up for clarity are denoted with a white asterisk. **b** Regions from of the cell edge from (**a**) were divided into windows for kymographic analysis of cell edge velocity and T-Plastin enrichment. Arrows depict edge velocities for each window between frames. **c** Kymographs of window edge velocities with protrusions shown in red (positive, $+0.65\,\mu m/s$) and retractions in blue (negative, $-0.65\,\mu m/s$). Parula kymograph shows the corresponding ratio of T-Plastin over F-tractin at same scale as (**a**) in the same windows analyzed for edge velocity. Dotted box shows region blown up in (**d**). **d** Region from (**c**) with protrusion areas highlighted in black and corresponding regions of T-Plastin enrichment also highlighted. **e** Temporal cross-correlation analysis of protrusion events shown in (**a–d**). Cross-correlation coefficients of T-Plastin (purple) and F-tractin (green) are shown relative the incidence of protrusion at 0 min. **f** Similar to (**a**), except cells also expressed MYL9-mTurq to mark myosin activity. Cells were treated with CK666 where indicated. **g** Similar to (**f**), except cells were first treated with 300 nM Jasplakinolide then CK666. Bars, 5 μm.

the KO cells spread significantly less compared to WT cells (Fig. 5c, 4 μm and 8 μm gaps). In contrast, when added to continuous fibronectin stripes without gaps, T-Plastin KO cells showed no reduction but instead a small increase in their cell spreading areas compared to WT cells (Fig. 5c, solid).

To determine if T-Plastin's role in facilitating gap bridging is unique to endothelial cells, we generated another CRISPR/Cas9 T-Plastin KO line in mesenchymal BJ-5ta fibroblasts[35] (Fig. S5a). Indeed, loss of T-Plastin also reduced the ECM gap bridging ability in BJ-5ta cells (Fig. S5b).

To understand when during the spreading process cells fail to bridge ECM gaps, we compared the relative change in cell area over time between WT and the average of all KO clones (Fig. 5d). On solid stripes, KO cells showed a small increase in their spreading rates compared to WT, suggesting that T-Plastin activity is dispensable for spreading on uniform ECM. Interestingly, on ECM gaps of 4 and 8 μm, KO cells showed similar initial rates to WT, but then diverged to become slower than WT after about 10 or 20 min, respectively (Fig. 5d). Together, this indicates that the initial adhesion and spreading ability of WT and KO cells are similar but the spreading rates of KO cells on gapped ECM become reduced compared to WT after cells have spread over a larger area. This suggests that ECM gap bridging defect arises when membrane tension forces start to increase, as has been shown when cells spread over an increasing area[36].

We next tested if membrane tension indeed has such a critical role in limiting protrusions across gaps in ECM. The amount of cholesterol in the plasma membrane is important for membrane fluidity[37] and sequestration of membrane cholesterol by the drug methyl-β-cyclodextrin (MbCD) has been shown to increase plasma membrane tension[38]. We, therefore, hypothesized that cells with increased membrane tension would be more resistant to protrusive actin forces and would inhibit a cell's ability to protrude across gaps in ECM. Indeed, when we pre-treated cells with 3 mM MbCD before adding the cells to the fibronectin ladder patterns we found that the initial increase in protrusion after ~10 min was dramatically reduced compared to untreated cells (Fig. S4h). Further, WT cells treated with MbCD showed a decreased ability to spread across gaps of both 4 and 8 μm compared to untreated cells (Fig. 5e). Strikingly, T-Plastin KO cells had a reduced capacity to bridge gaps compared to WT, but the remaining capacity to spread was unaffected by MbCD treatment (Fig. 5e). Thus, increasing membrane tension reduces protrusion in WT cells but has no significant effect in KO cells likely because KO cells are already deficient in protrusion at a lower membrane tension without being further affected by MbCD treatment.

To further test the hypothesis that membrane tension opposes the protrusive role of T-Plastin, we reduced the force exerted by membrane tension by adding deionized water to generate hypoosmotic conditions that pushes membranes away from the

underlying actin network and is known to increase membrane protrusions in HUVEC and other cell types[39–41] (Movie S8). To test this during ECM gap bridging, we added WT and T-Plastin KO HUVEC to fibronectin ladder patterns with 8 μm gaps and allowed cells to spread for at least 30 min, a time where both WT and KO cells showed a plateau in their ability to spread (Fig. 5d, 8 μm gaps). Indeed, addition of deionized water to cells already spread on ladder patterns resulted in a significant increase in the ability of both WT and KO cells to make contact with additional fibronectin stripes (Fig. 5f, ddH$_2$O), suggesting that membrane tension restricts protrusion in WT and KO cells.

As controls, we took a similar approach but instead added either control buffer or the Arp2/3 inhibitor CK666 and measured the change in the number of additional fibronectin stripes that cells were able to reach. As expected, since most cell areas had already plateaued, there was no significant difference in the number of fibronectin stripes that cells occupied upon inhibition of Arp2/3 (Fig. 5f, control and CK666). Together, these results support the hypothesis that T-Plastin has a critical role in strengthening the actin network to promote protrusion under conditions where membrane tension becomes limiting.

We next examined how myosin contractility relates to the T-Plastin-mediated regulation of protrusion by inhibiting the actomyosin contractile machinery with treatment of a combination of 3 μM blebbistatin and 20 μM of the ROCK inhibitor Y-27632. Interestingly, both WT and KO cells were able to further protrude and make contact with additional fibronectin stripes to a similar degree, suggesting that myosin-mediated contraction can restrict ECM gap bridging by a mechanism that is largely independent of T-Plastin activity (Fig. 5f, Bleb/Y27).

We next tested the role of focal adhesions during the ECM gap bridging process. Since T-Plastin and nascent adhesions both have positive roles in protrusion, we considered that the two processes are likely synergistic with each other during ECM gap bridging. To determine if there is an epistatic relationship between T-Plastin facilitated protrusion and focal adhesions, we lowered adhesion strength by using sub-maximal doses of an adhesion inhibitor peptide (cRGD) and measured the gap-bridging abilities of WT and KO HUVECs. Before adding cells to fibronectin ladder patterns, we added 100 μM of cRGD, a concentration that enhances cell velocity in more traditional wound healing assays, where migration speed is reduced by high relative ECM adhesion (Fig. S4i). We found that on gaps of 4 μm, WT cells showed a significant reduction in spreading when treated with cRGD compared to control cells, suggesting that failed adhesion to new fibronectin rungs is limiting the gap bridging process (Fig. 5g, left). Interestingly, KO cells treated with the same concentration of cRGD showed no significant effect on gap bridging efficiency for 4 μm gaps. Additionally, on larger gaps of 8 μm, treatment with cRGD markedly again reduced the spreading area of WT cells but the same treatment also had little

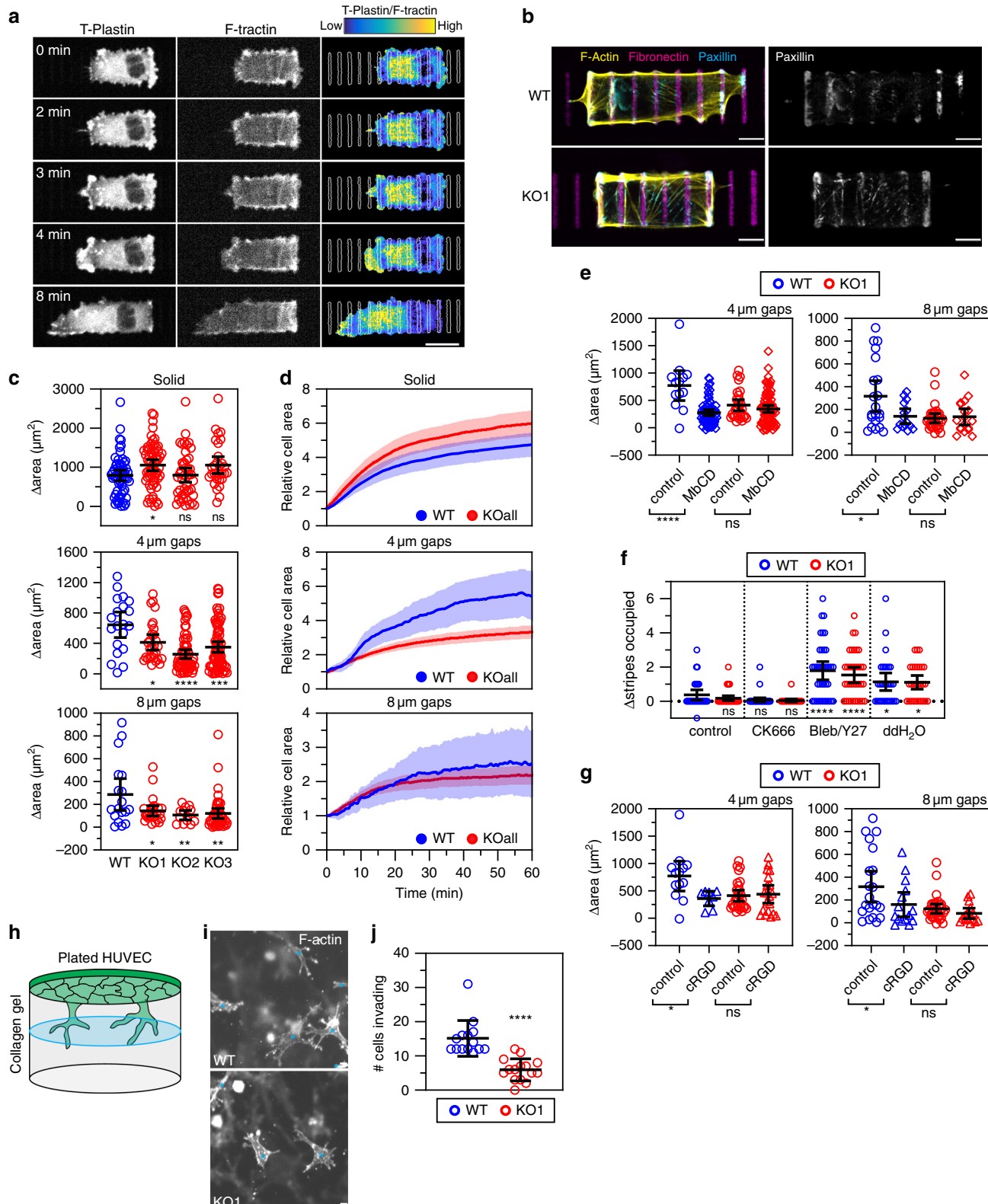

effect on KO cells (Fig. 5g, right). These results are consistent with a model that the two processes work in an epistatic manner: Integrin adhesion becomes more rate-limiting for forming new adhesion contacts after a T-Plastin-driven protrusion crosses a several-micron wide ECM gap.

Altogether, our different lines of experiments show that membrane protrusion across ECM gaps can be restricted by (i) insufficient T-Plastin-mediated strengthening of the actin network, (ii) increased membrane tension that pushes the front membrane against the protrusive actin network, (iii) weakened ECM adhesion, and (iv) increased actomyosin contractility. Furthermore, these results demonstrate that T-Plastin is synergistically required along with adhesion to drive cellular protrusions across gaps in ECM under conditions of increased membrane tension.

**Fig. 5 T-Plastin promotes cell protrusions across gaps in ECM. a** HUVEC expressing T-Plastin-mRuby3 and F-tractin-mCitrine on ladder patterns with 4 µm gaps, bar 20 µm. T-Plastin enrichment over F-tractin is shown as a parula colormap (yellow and blue are 99th (high), 3rd (low) percentiles). **b** Wildtype (WT) and T-Plastin KO1 HUVEC on fibronectin micropatterns were stained for F-actin (yellow) and paxillin (cyan), bars 10 µm. **c** 3 HUVEC T-Plastin KO lines (KO1-3) (red) change in area 1 h after first contact on a stripe with different sized gaps compared to WT (blue). (*n* cells for Solid are: WT = 59, KO1 = 62, KO2 = 45, KO3 = 30; 4 µm gaps: WT = 20, KO1 = 29, KO2 = 61, KO3 = 74; 8 µm gaps: WT = 18, KO1 = 26, KO2 = 13, KO3 = 45; taken from ≥3 independent replicates). **d** Relative cell area changes over a 1 h period in both WT (blue) and all KO lines (red) used in (**c**). Means are shown as solid lines, transparent regions indicate the 95% confidence intervals. **e** WT and T-Plastin KO1 HUVEC were treated with MbCD (*n* cells for WT and KO are: control 4 µm gaps 13 and 29, MbCD 4 µm 68 and 82, control 8 µm 21 and 31, MbCD 8 µm 15 and 18, respectively; controls are same from (**g**); taken from ≥3 independent replicates). **f** WT and KO1 HUVEC were pre-incubated on 8 µm gap patterns for 2 h prior to the indicated treatment. The change in the number of fibronectin stripes contacted 45 min after treatment is shown (n cells for WT and KO1 are: control 30 and 40, CK666 38 and 24, Bleb/Y27 38 and 36, ddH$_2$O 30 and 28, respectively; taken from ≥3 independent replicates). Each condition was compared to WT control. **g** Similar to (**e**), but treated with cRGD (*n* cells for WT and KO are: control 4 µm gaps 13 and 29, cRGD 4 µm 8 and 20, control 8 µm 21 and 31, cRGD 8 µm 15 and 15, respectively; controls are same from (**e**); taken from ≥3 independent replicates). **h** 3D invasion diagram. HUVEC are plated on top of collagen gel and invade. **i** WT and KO1 HUVEC stained for F-actin. Invading cells are marked by a cyan asterisk. Bars, 10 µm. **j** Quantification of collagen invasion. Black bars represent mean and SD. (*n* slide wells for WT = 14 and KO1 = 15; taken from three independent replicates). Unless indicated elsewhere, black bars represent the mean and 95% confidence intervals, *$P < 0.05$, **$P < 0.01$, ***$P < 0.001$, ****$P < 0.0001$, ns = not significant, analyzed using a one-way ANOVA with Sidak's or Dunnett's multiple comparison test for (**c**, **e**, **f**, **g**) or with an two-tailed unpaired *t*-test for (**j**).

**Functional consequences of T-Plastin deficiency**. We next tested whether the observed defects in lamellipodial structure and protrusion in T-Plastin deficient cells result in cell migration defects. To monitor cell migration into an open space, we used a broadly established cell migration assay where a Delrin-tip scratch tool is used to remove a stripe of cells (Fig. S6a)[42]. In this assay, the scratch also removes much of the ECM, creating areas devoid of ECM that cells must traverse and likely bridge or remodel as they migrate (Fig. S6a, b). Using this assay, we tracked the velocity of individual cells for 14.5 h as they migrated into the open space using a nuclear Hoechst stain (Fig. S6c). Knockdown of T-Plastin caused a dramatic reduction in cell velocity and significantly reduced the ability of the monolayer to fill the scratch area compared to control siRNA treated cells (Fig. S6d, e). We next performed the same scratch-based migration assay using T-Plastin KO HUVEC clones, comparing them to a parental line expressing only Cas9. Again, all three KO lines showed reduced cell velocities compared to the control Cas9 expressing cells (Fig. S6f). Thus, the role of T-Plastin in promoting membrane protrusions is functionally important for directed cell migration.

To determine if the findings using fibronectin ladder patterns are representative of what occurs in more physiological three-dimensional (3D) ECM matrices, we performed invasion assays of WT and T-Plastin KO HUVEC into 3D collagen gels[43] (Fig. 5h). Cells were seeded on top of 2.4 mg/mL collagen gel in angiogenesis chambers. After 24 h, cells were stained for F-actin, and the number of cells that invaded into the collagen gel were quantified (Fig. 5i, j). The ability of T-Plastin KO cells to invade the collagen gel was markedly reduced compared to WT, indicating that the defects seen with ECM gap bridging and in the scratch assay also apply to 3D migration models representing a more physiological ECM environment.

**T-Plastin broadens and lengthens protrusions without affecting actin treadmilling rates or focal adhesion dynamics**. We next examined lamellipodia in T-Plastin KO cells to gain further insight into T-Plastin's role in facilitating protrusion. First, we examined the width of protrusion edges in sparsely plated HUVEC using endogenous Arp3 as a marker of active protrusive zones (Fig. 6a). WT cells showed broad Arp3-based protrusions with an average width of ~32 µm while KO cells had narrower protrusions of ~19 µm (Fig. 6a, b). This reduction in protrusion width was partially rescued by stable re-expression of a CRISPR-resistant T-Plastin. This suggests that T-Plastin strengthens the branched actin network and allows protrusions to reach a critical size both in width and length.

One possible explanation for protrusion defects in the absence of T-Plastin could be that the protrusive actin network itself becomes more unstable without an actin-bundling protein, and therefore exhibits a faster turnover (or treadmilling) than the actin network of WT cells. To directly test this possibility, we generated WT and T-Plastin KO HUVEC expressing stoichiometric levels of F-tractin-mRuby3 and photoactivatable-GFP (PAGFP)-actin[44,45]. This allows for simultaneous visualization of F-actin and quantification of actin subunit turnover within a region of interest after subsequent photoactivation with low intensity 405 nm light. We photoactivated small regions in protrusions of sparsely plated WT and KO HUVEC and quantified the intensity of PAGFP-actin in regions that carefully encompassed the protrusion as it moved outward but also avoiding signals from mature focal adhesions/ends of stress fibers to ensure we only capture the turnover of the protrusive actin network at the very leading edge (Fig. 6d). Markedly, there was no significant difference in actin turnover in WT and KO cells, with both showing a reduction of the PAGFP-actin fluorescence signal to about 50% after ~20 s (Fig. 6c). As a control, we treated cells with a cocktail of 20 µM Y-27632, 8 µM Jasplakinolide, and 5 µM Latrunculin-B (JLY) to 'freeze' the actin cytoskeleton[46] and observed the expected persistent PAGFP-actin signal that failed to decay over longer periods of time (Fig. 6c). These results suggest that the protrusion defects seen in T-Plastin KO cells are not a result of accelerated actin turnover at the leading edge.

To further evaluate a potential role of T-Plastin in regulating adhesion, we examined several classical markers of adhesion in WT and KO HUVEC. We first stained for F-actin and paxillin in protrusions of HUVEC plated on continuous collagen (Fig. 6e). There was no overt difference in the size or number of adhesions formed between WT and KO cells, similar to our observations in cells on the fibronectin ladder patterns (Figs. 5b and S5e). However, in typical WT HUVEC protrusions there is a separation of ~1–5 µm between the outermost edge marked by smaller nascent adhesions and the back end of lamellipodia where larger mature focal adhesions are present. This separation was reduced in all HUVEC T-Plastin KO clones, and larger mature adhesions could be found very close to the smaller nascent adhesions at the outer edge, suggesting that T-Plastin regulates the length of protrusions by increasing the distance between nascent and mature adhesions sites (Fig. 6e, cyan and yellow lines, respectively).

In an additional control, we examined whether T-Plastin may affect focal adhesion assembly or disassembly by measuring focal adhesion kinetics in WT and KO HUVEC expressing

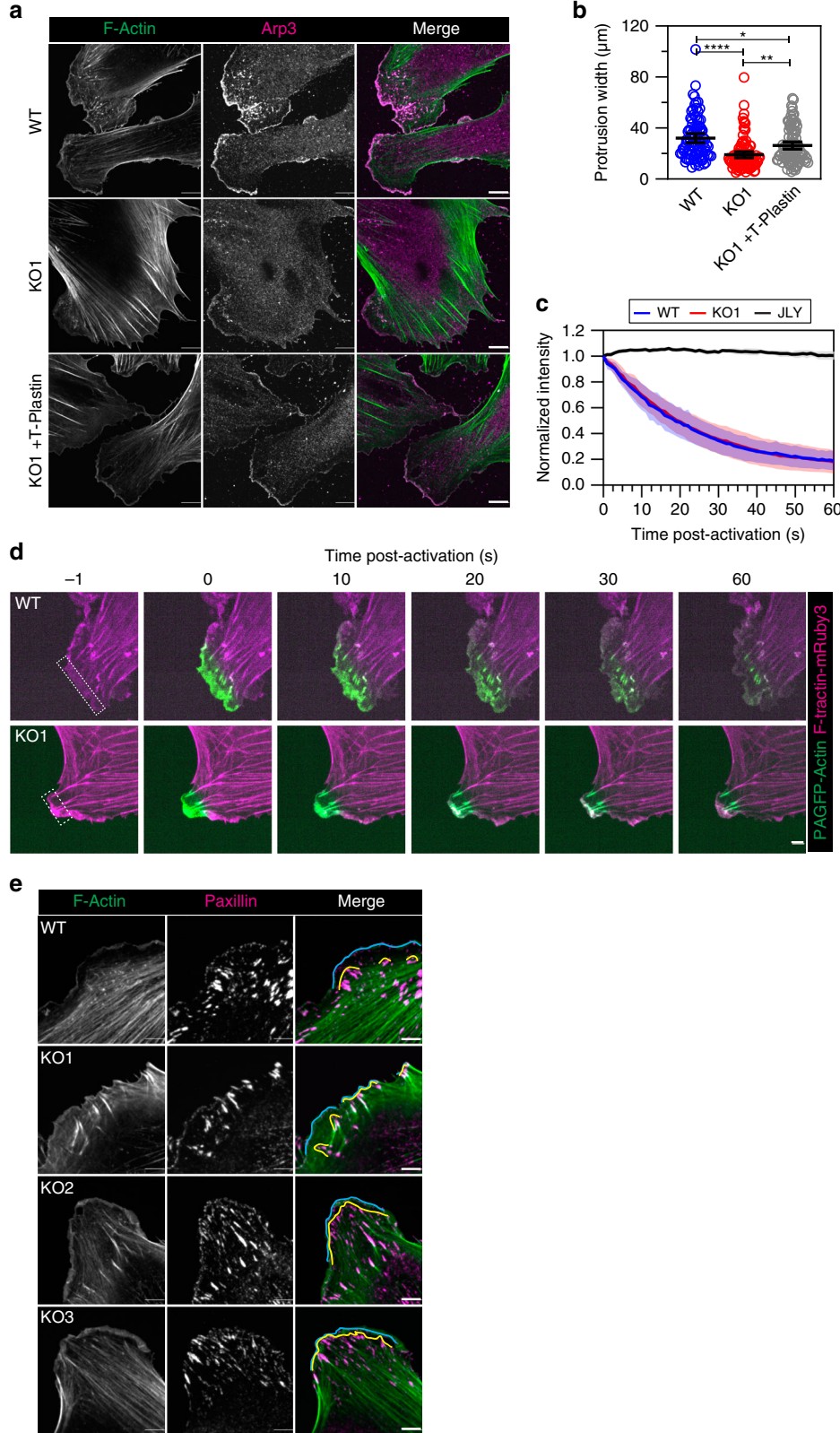

F-tractin-mCitrine and mCherry-Paxillin. Using a previously characterized focal adhesion analysis tool[47,48], we found no significant difference in either the assembly or disassembly rates of focal adhesions between WT and T-Platin KO cells (Fig. S7a). Additionally, the distribution of nascent and mature adhesions in live WT and KO cells did not show any overt differences

(Fig. S7b, c). These results support the hypothesis that T-Plastin primarily regulates the strength and size of protrusions but not adhesion itself.

Taken together, the different lines of evidence suggest that T-Plastin helps protrusions reach a critical size—by increasing both the overall width and the distance between nascent adhesions and

**Fig. 6 Loss of T-Plastin reduces protrusion width and length but has no effect on actin turnover. a** Sparsely plated WT, KO1, and KO1 HUVEC stably expressing a CRISPR-resistant T-Plastin (KO1 + T-Plastin) stained for F-actin and Arp3. Bars. 10 μm. **b** Quantification of protrusion widths based on their Arp3 staining in cells as shown in (**a**). Black bars indicate the mean and 95% confidence intervals for WT (blue, n protrusions = 88), KO1 (red, n protrusions = 101), KO1 + T-Plastin (gray, n protrusions = 96) taken from two independent replicates. *$P < 0.05$, **$P < 0.01$, ****$P < 0.0001$, analyzed using a one-way ANOVA with Tukey's multiple comparison test. **c** Quantification of photoactivated protrusive actin intensities in WT (blue, n cells = 14) and KO1 (red, n cells = 17) cells as shown in (**d**). As a control, cells were treated with a mixture of Jasplakinolide, Latrunculin-B, and Y-27632 (JLY, black, n cells = 8); taken from three independent replicates. Means are shown as solid lines, transparent regions indicate 95% confidence intervals. **d** Photoactivation images of WT and KO1 HUVEC expressing PAGFP-Actin (green) and F-tractin-mRuby3 (magenta). A small region of the protrusion was activated and measured (white boxes). Bar, 5 μm. **e** Sparsely plated WT and T-Plastin KO cell lines (KO1-3) were fixed and stained for F-actin (green) and paxillin (magenta). The areas containing nascent and mature focal adhesions are marked by cyan and yellow lines, respectively. Bars, 5 μm.

larger mature focal adhesions—and that this increase in size is achieved not through alteration of actin or adhesion turnover, but through the strengthening of the protrusive actin network against the opposing force of membrane tension.

**Ultrastructural analysis reveals that loss of T-Plastin reduces the resilience of the protrusive actin network.** To directly visualize the ultrastructural cytoskeletal properties of the protrusion defects in T-Plastin KO cells, we processed and examined sparsely plated HUVEC with platinum replica electron microscopy (PREM)[49,50] (Fig. 7a). WT cells showed typical broad protrusions consisting of a thick branched actin network at the leading edge, however, this actin network was often much thinner in KO cells. The distance that these branched actin networks extended from the leading edge of protrusions averaged ~2 μm and 900 nm in WT and KO cells, respectively (Fig. 7b). Interestingly, when inspecting the images at high magnification, the actin filaments at the leading edge did not show overtly different geometries or arrangements. Rather, the density of actin filaments appeared slightly reduced, consistent with decreased phalloidin staining seen via immunofluorescence imaging (Fig. 7a bottom and 2c). Additionally, WT cells showed typical straight filopodia emanating from broader lamellipodial protrusions, but in KO cells, these filopodia were more often bent (Fig. S8a, arrows). Together, these results are consistent with the interpretation that the protrusive actin networks of filopodia and lamellipodia both being less resilient to forces of membrane tension in the absence of T-Plastin's bundling activity, causing more narrow membrane protrusions that are unable to reach as far.

To better understand the role of T-Plastin in strengthening the protrusive actin network we also examined it's ultrastructural localization with stimulated emission depletion (STED) super-resolution microscopy[51] in HUVEC. T-Plastin was highly enriched in all actin structures at the leading edge of protrusions, areas consistent with both linear bundles in filopodia and branched actin in the Arp2/3-generated lamellipodial network (Fig. 7c). Examination of T-Plastin KO HUVEC also showed a reduction in the protrusive actin zone between the leading edge and ends of stress fibers in the lamellum (Fig. 7c, yellow lines). Our two-color STED microscopy showed an apparent colocalization near the cell edge (blue inset) and to a lesser degree outside of the leading edge of the protrusion (orange inset). We analyzed the colocalization of F-actin and T-Plastin in regions corresponding to either protrusions or cytoplasmic areas outside of the protrusion (regions defined in Fig. S8b) by calculating the intensity correlation quotient (ICQ)[52]. We found that T-Plastin colocalizes with F-actin within the protrusions, as shown by positive ICQ values, but has a random distribution relative to actin in the areas outside of the protrusions, as shown by the overlap in the ICQ values with the scrambled-signal controls (Fig. 7d). These results confirm with high spatial resolution that T-Plastin localizes to both filopodial and

lamellipodial actin structures. Together with the actin cytoskeleton defects in both lamellipodia (Fig. 7a) and filopodia (Fig. S8a), our data suggests that T-Plastin plays an important structural role in both types of protrusions regardless of differences in actin filament geometry.

**Discussion**

Here we use micropatterns of ECM with gaps containing non-adherent PLL-PEG to investigate how membranes can protrude and bridge ECM gaps where cells cannot form adhesions. There are three fundamental parts to this gap bridging process: (i) membrane tension which limits the actin polymerization-driven membrane protrusion, (ii) the strength of ECM adhesions for the actin cytoskeleton to push against to start a protrusion and to reattach after an ECM gap has been bridged, and (iii) the mechanical strength of the actin network between the leading membrane edge and the last ECM adhesion sites. We show that the ancient actin-bundling protein, T-Plastin plays a critical role for cells to protrude across such gaps in ECM. Specifically, we show that T-Plastin works synergistically with focal contacts to strengthen protrusive actin networks at the leading edge and enables cells to protrude across ECM gaps under conditions of high membrane tension (Fig. 8).

Mechanistically, our study shows that T-Plastin localizes to linear and branched actin networks in lamellipodia and is critical to maintain their structure. T-Plastin is selectively enriched in active protrusions and becomes depleted prior to retraction events. Furthermore, T-Plastin localization shows a marked anti-correlation with regions of myosin contractile activity, in support of a selective role of T-Plastin in reinforcing membrane protrusions and preventing myosin II-based contraction. One plausible mechanism for this selectivity is that actomyosin bundled filaments are spaced too far from each other to allow for stable binding of a compact bundler such as T-Plastin, which would be unable to bind to both filaments simultaneously. Indeed, such a spatial selectivity has been shown in vitro where α-actinin bundled filaments preclude fimbrin or fascin bundling activities[34]. Interestingly the fission yeast plastin Fim1 can also displace the actin-binding protein tropomyosin Cdc8[53], suggesting such competition and exclusivity mechanisms have ancient origins and can be highly conserved.

Interestingly, T-Plastin localizes to multiple protrusive actin structures such as filopodia, which contain tight parallel bundles of actin filaments, and broader lamellipodial structures, which contain a dense network of branched actin (Fig. 7a, c). Fascin, another compact bundler, shows exclusivity to these tight parallel bundles[14], suggesting that T-Plastin is either more promiscuous in its bundling geometries, or that similarly spaced pairs of parallel actin filaments exist within this dense branched actin network to a degree high enough to account for T-Plastin's remarkably specific localization. Further structural and reconstitution studies are needed to better explore the types of bundle geometries that T-Plastin can generate or associate with.

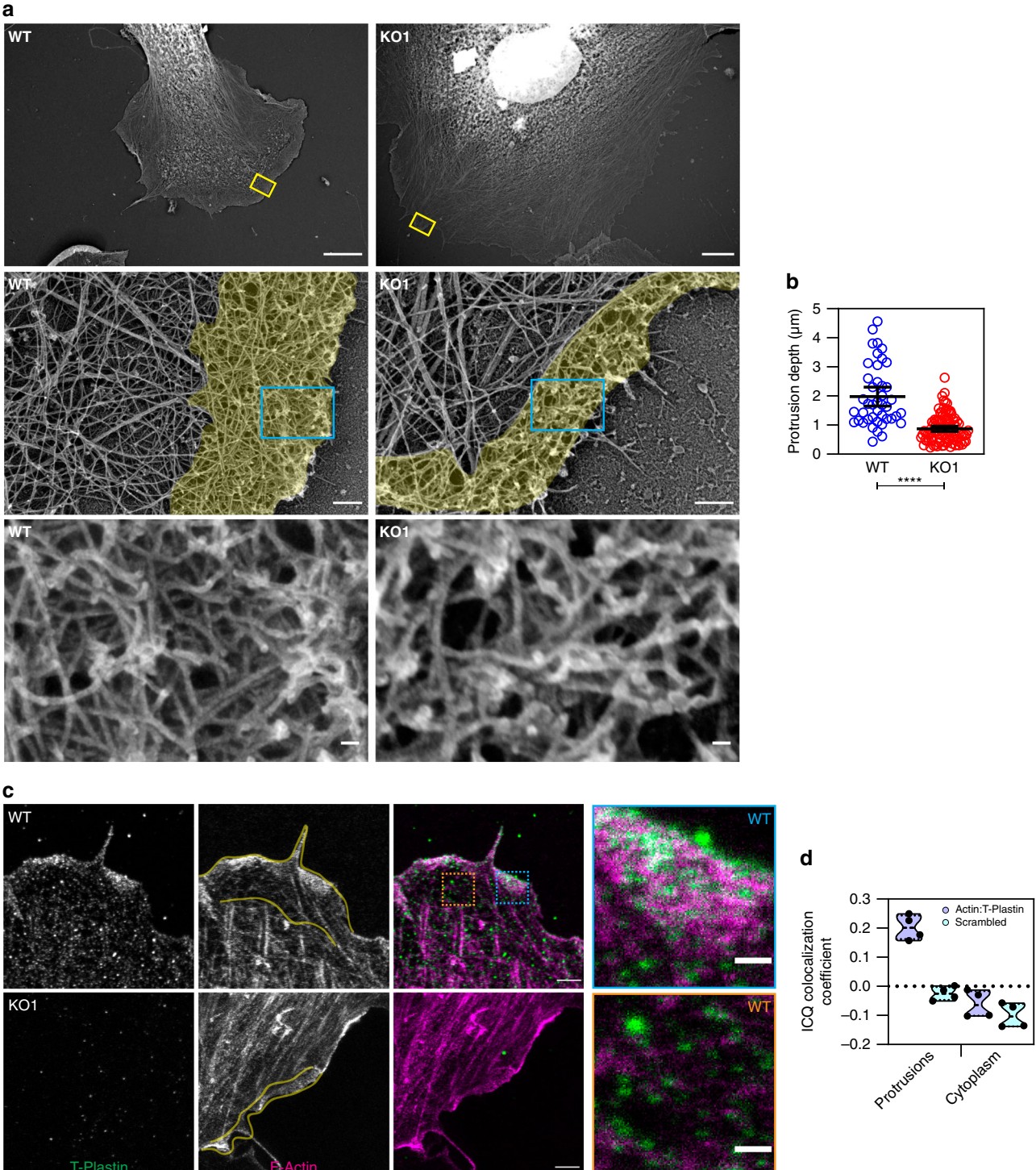

**Fig. 7 Ultrastructural analysis showing that T-Plastin localizes to and strengthens protrusive actin structures. a** PREM images of WT and T-Plastin KO1 HUVEC protrusions. Top, low magnification images of cell protrusions, regions in yellow boxes correspond to magnified regions shown in middle. Bars, 5 μm. Middle, magnified areas from top showing the leading edge. Cyan boxes correspond to the further magnified regions shown in bottom. The depth of the branched protrusive network is highlighted in yellow. Bars, 200 nm. Bottom, Highly magnified areas from middle showing the arrangement of actin filaments at the very leading edge. Bars, 25 nm. **b** Quantification of protrusion depth, the distance the branched actin network persists from the leading edge of protrusions. Black bars represent the mean and 95% confidence intervals, analyzed using unpaired t-test, **** P < 0.0001 (*n* protrusions WT = 43, KO = 100; taken from ≥3 independent replicates). **c** STED super-resolution images of WT and KO1 HUVEC stained for T-Plastin (green) and F-actin (magenta). Regions of high and low actin density in a WT protrusion are magnified in orange and cyan boxes as indicated. Bars 2 μm and 500 nm, respectively. Yellow lines indicate extent of the protrusion. **d** Violin plot of intensity correlation quotients between T-Plastin and F-actin (purple), quantified in protrusion and cytoplasmic regions (see Fig. S8b, for example). As a control (cyan) the values for F-actin were scrambled. Black dots indicate individual values for different images (*n* cells = 4, representative of three independent replicates), medians are shown as black dotted line within colored violins.

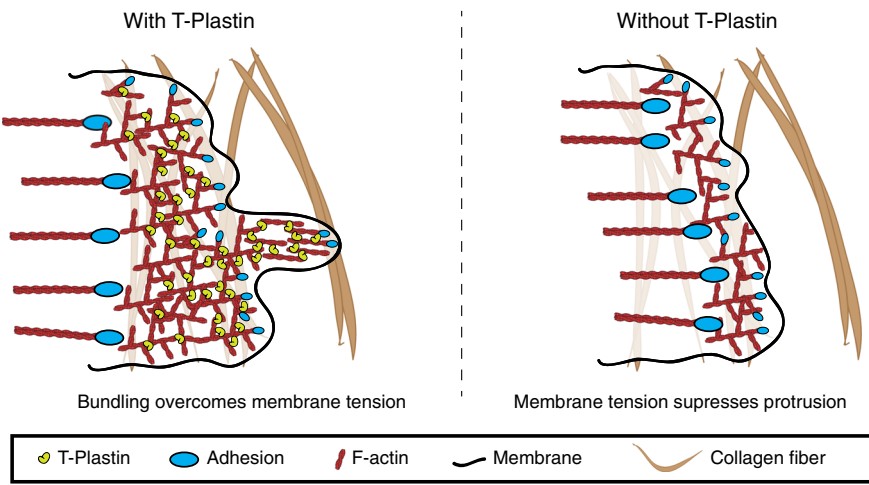

**Fig. 8 Model of how T-Plastin promotes ECM gap bridging.** Left, T-Plastin strengthens and extends the protrusive actin network between nascent adhesions at the leading edge and mature focal adhesions further back in the lamellum. This reinforcement facilitates a cell's ability to protrude across gaps in the ECM. Right, in the absence of T-Plastin cells are unable to overcome the opposing force of membrane tension, the protrusive actin network becomes reduced in size, and cells exhibit an inability to protrude across ECM gaps.

Myosin, α-actinin, and the plastins represent the most ancient actin-bundling proteins, being conserved in distant eukaryotes such as plants. Non-muscle myosin II and α-actinin both play important roles in the assembly and function of contractile actin networks, while we here show that T-Plastin is critical for the stabilization of protrusive actin. In support of such a role for other members of the plastin family, the T-Plastin homolog L-Plastin is selectively expressed in hematopoietic cells[54] but is also highly upregulated in many cancer types as a strong promoter of metastasis[55]. This is likely a consequence of plastin actin-bundling activity promoting effective migration across a diverse ECM landscape in vivo during metastasis.

## Methods

**Cell culture**. All experiments using HUVEC were performed with a hTERT-immortalized HUVEC line previously described[32]. All HUVEC and BJ-5ta stably expressing fluorescent reporters were generated from this hTERT-immortalized line by lentiviral transduction followed by either FACS or antibiotic selection (0.5 μg/ml puromycin, 10 μg/ml blasticidin, or 0.5 mg/ml neomycin). T-Plastin KO HUVEC clones were sequentially transduced with lentivirus and selected with blasticidin and puromycin. Single cell-derived clones were then generated by serial dilution into 96-well plates and T-Plastin knockout was determined by immunofluorescence and western blot against endogenous T-Plastin. HUVEC were cultured in either EGM2 (Lonza) or EndoGRO VEGF complete media (Millipore). HEK 293T cells for generating lentivirus were maintained in DMEM supplemented with 10% FBS and 1% GlutaMAX (Thermo Fisher Scientific). hTERT-RPE-1 and B16-F1 cells (ATCC) were maintained in DMEM-F12 or DMEM with 10% FBS (Thermo Fisher Scientific), respectively. BJ-5ta cells were maintained in 4:1 DMEM:M199 with 10% FBS (Thermo Fisher Scientific). T-Plastin KO BJ-5ta clones were concurrently transduced with lentivirus and selected with blasticidin and puromycin. Single cell-derived clones were then generated by serial dilution into 96-well plates and T-Plastin knockout was determined by immunofluorescence. The absence of mycoplasma contamination from all cell cultures was routinely verified using a PCR test. Cultured cell diameter and number were quantified using a coulter-method-based cytometer (Scepter, Millipore).

**Antibodies and reagents**. Phalloidin conjugated to Alexa Fluor 488 (A12379, used at 1:500 for immunofluorescence (IF)), secondary antibodies against rabbit or mouse IgG conjugated to Alexa Fluor 568 or 647 (A11036, A11004, A32733, A21235, all used at 1:500 (IF)), and Hoechst 33342 (H3570, used at 1:50,000) were from Thermo Fisher Scientific. Mouse monoclonal and rabbit polyclonal antibodies against T-Plastin were from Novus (5B9, used at 1:100 (IF) and 1:1000 for western blot (WB)) and Thermo Fisher Scientific (PA5-27883, used at 1:100 (IF) and 1:000 (WB)) respectively. The rabbit monoclonal antibody against paxillin was from Abcam (ab32084, used at 1:200 (IF)). The mouse monoclonal antibodies against α-Tubulin (T5168, used at 1:5000 (WB)) and Arp3 (A5979, used at 1:250 (IF)) were

from Sigma. The rabbit monoclonal antibody against GAPDH was from Cell Signaling (5174, used at 1:3000 (WB)). For STED imaging, the STAR 635P-labeled phalloidin (ST635P, used at 1:25 (IF)) and STAR 520SXP-labeled goat anti-mouse secondary antibody (ST520SXP, used at 1:250 (IF)) were from Abberior. For western blotting secondary antibodies against rabbit or mouse IgG conjugated to IRDye 680LT (926-68021, used at 1:10,000(WB)) or IRDye 800CW (926-32210, used at 1:10,000 (WB)) were from LI-COR Biosciences. CK666 (SML0006), methyl-β-cyclodextrin (C4555), Blebbistatin (B0560), and cylic-RGD cell adhesion inhibitor peptide (SCP0111) were from Sigma, Jasplakinolide was from Cayman Chemical (11705), ML-7 (sc-200557), and Y-27632 (sc-281642) were from Santa Cruz Biotechnology, Latrunculin-B was from Abcam (ab144291), and bFGF was from R&D Systems (223-FB). Bovine collagen was from Advanced BioMatrix (5005-100 ML). Bovine fibronectin was from Sigma (F1141).

**DNA constructs and lentivirus production**. HUVEC stably expressing the stoichiometric myosin II/F-actin activity reporter (F-tractin-mCherry-p2a-MYL9-mTurquoise) were described previously[32]. pLV-Ftractin-mCitrine was generated by PCR amplification of Ftractin-mCitrine and Gibson assembly[56] into pLenti-EF1a-MCS. pLV-RaichuCdc42-IRES-Blast was based on RaichuCdc42[23], but for the FRET pair YPet/mTurquoise, YPet was replaced by a codon-changed version of YPet, optimized to be dissimilar to mTuruqoise, in order to avoid recombination between the sequences encoding the fluorescent proteins during lentiviral transduction[57]. After subcloning, the final codon optimized RaichuCdc42 was inserted via Gibson assembly into pLV-EF1a-MCS-IRES-Blast[32] (Addgene # 85133).

Human T-Plastin in pDONR223 (PLS3, human ORFeome v5.1) was used to generate T-Plastin-mCitrine by LR recombination using the custom-made Gateway destination vector pmCitrine-N-DEST/TO. pLV-T-Plastin-mRuby3-IRES-Blast was made by Gibson assembly of PCR products of T-Plastin and mRuby3 and entry into pLV-EF1a-MCS-IRES-Blast. pLV-T-Plastin-mRuby3-p2a-MYL9-mTurquoise-IRES-Blast was made by removing the F-tractin-mCherry from pLV-F-tractin-mCherry-p2a-MYL-mTurquoise-IRES-Blast with BamHI and inserting PCR-amplified T-Plastin-mRuby3 via Gibson assembly. pLV-F-tractin-mRuby3-p2a-PAGFP-βactin-IRES-Neo was made by inserting PCR-amplified regions of PAGFP, human β-actin, and F-tractin-mRuby3 via Gibson assembly.

EGFP-tagged supervillin, Eplin-α and β were gifts from Elizabeth Luna (Addgene plasmids 13040, 40947 and 40948, respectively)[58]. α-Actinin1-GFP was a gift from Carol Otey (Addgene plasmid 11908)[59]. mCherry-Paxillin was a gift from J. Victor Small[60]. mEmerald-Fascin and Filamin-A were gifts from Michael Davidson (Addgene plasmids 54095 and 54098 respectively). Lenti-Cas9-Blast, and LentiGuide-Puro were gifts from Feng Zhang. T-Plastin sgRNA guide sequences were designed using cripsr.mit.edu and assembled as previously described[61].

Lentivirus was produced in HEK293T cells co-transfected with a third-generation packaging plasmid mixture (pMDLg/pRRE, pRSV-rev, pCMV-VSVG) and a transfer vector containing the gene of interest using Lipofectamine 2000 (Thermo Fisher Scientific)[62]. Viral supernatants from 48 hr and 72 hr post transfection were pooled, 0.22 μm filtered, and concentrated using centrifugal filters with a 100 kDa cutoff (Millipore).

To generate CRISPR KO T-Plastin cell lines sgRNA sequences targeting T-Plastin (Fig. S4A) were ligated into lentiGuide-Puro plasmid[61] cut with BsmBI (New England Biolabs). Cells were transduced with lentiCas9-Blast and

sgRNA-containing virus as described above and as described previously[61]. All primers used in this study are listed in Supplementary Table 1.

**Transient transfections of DNA and siRNA.** For transient transfections of DNA, $1.5 \times 10^4$ HUVEC, or $5 \times 10^3$ RPE-1 or B16-F1 were plated the day before transfection in a glass-bottom 96-well plate (Cellvis) coated with 31 μg/mL collagen. The day of transfection, culture medium was replaced with 80 μL of antibiotics-free culture media per well. Then, 0.2 μg DNA and 0.25 μL Lipofectamine2000 (Thermo Fisher Scientific), mixed in 20 μL OptiMEM (Thermo Fisher Scientific), was added following the manufacturer's instructions. This transfection mix was replaced after 3 h with culture media and cells were imaged after 24 h.

Pooled and individual siRNA targeting human T-Plastin were from Dharmacon. For transfection, $1 \times 10^4$ HUVEC per well were suspended in 80 μL of antibiotics-free EGM2, and reverse-transfected with siRNAs diluted in 20 μL OptiMEM to a final 20 nM concentration with 0.25 μL of Lipofectamine RNAiMAX (Thermo Fisher Scientific). The transfection mix was replaced with EGM2 after 8hrs. At 48hrs post transfection cells were analyzed.

**Western blotting.** HUVEC grown in 60 mm dishes were lysed in 25 mM Tris pH 7.4, 100 mM NaCl, 1%TritonX-100, 1 mM EDTA with Halt Protease Inhibitor (Thermo Fisher Scientific), concentrations were measured with Pierce 660 nm Protein Assay Kit (Thermo Fisher Scientific), and then boiled in SDS sample buffer. For samples treated with siRNA, the number of cells and transfection mix were scaled for the increased volume of the 60 mm dish. Protein samples were separated by SDS-PAGE and transferred to Immobilon-FL (Millipore). All Western blots were imaged using an Odyssey infrared imaging system controlled by ImageStudio software (LI-COR Biosciences).

**Fixed and live-cell imaging.** Live cells were imaged in extracellular buffer (125 mM NaCl, 5 mM KCl, 1.5 mM MgCl2, 1.5 mM CaCl2, 20 mM HEPES, supplemented with 10 mM D-glucose, 1% FBS, and 5 ng/mL bFGF. For fixed imaging, cells were rapidly fixed in 37 °C 4% paraformaldehyde in PBS, washed in PBS, permeabilized in 0.2% TritonX-100 in PBS, washed again and blocked in 3% FBS in PBS prior to primary and secondary antibody staining.

Images shown in Figs. S5a and S6a–c were acquired on a fully automated fluorescence microscope (ImageXpress Micro XL, Molecular Devices), equipped with a 20×0.75NA objective (Nikon), a Sola Light Engine (Lumencor), environmental control, a Zyla 5.5 sCMOS camera (Andor), and controlled by MetaExpress software (Molecular Devices).

Images shown in Figs. 2e and S2a and b were captured on a custom-built widefield fluorescence system consisting of a Zeiss Axiovert 200 M microscope enclosed in an environmental control chamber (Haison), ORCA Flash 4.0 LT sCMOS camera (Hamamatsu), 100 W HBO lamp, and 63× 1.2 NA water immersion and 40 × 1.3NA oil immersion objectives.

Unless described elsewhere, all other images were acquired on an automated confocal system controlled by Slidebook software (Intelligent Imaging Innovations, 3i). The system consists of an Eclipse-Ti stand with perfect focus system (Nikon Instruments), CSU-W1 spinning disc (Yokogawa), a 3i laser stack with 405, 445, 488, 515, 561, and 640 nm lines, an environmental chamber (Haison), 40× 1.3NA, 100× 1.4NA, and 60× 1.35NA oil immersion objectives and a ×60 1.27NA water immersion objective, and two Zyla 4.2 sCMOS cameras (Andor) and with motorized dichroics enabling simultaneous acquisition of channels.

**3D structured-illumination microscopy (3D-SIM).** HUVEC were plated as monolayers on 22 mm square coverglass (Zeiss) coated with 31 μg/mL of collagen. 24 h later a scratch was generated with a pipette tip, and then incubated for 4 h at 37 °C before fixation with 4% paraformaldehyde. After permeabilization and blocking with 3% FBS cells were stained for paxillin and F-actin, washed again in PBS, and mounted in Vectashield antifade (Vector Labs) for imaging. Images were acquired on an OMX BLAZE 3D-SIM microscope (GE Lifesciences) with 100×1.42NA objective and Evolve 512 delta cameras (Photometrics). Images were processed using SoftWoRx (GE Lifesciences) and Fiji[63].

**Photoactivation.** WT and T-Plastin KO HUVEC stably expressing F-tractin-mRuby3-p2a-PAGFP-βactin-IRES-Neo were imaged by spinning disc microscopy using 488 and 561 nm laser lines acquiring images every second. Photoactivation was achieved using low intensity of 405 nm laser light and rasterization across user-defined regions (Vector, Intelligent Imaging Innovations). Analysis regions were defined in Slidebook to exclude the ends of actin stress fibers/focal adhesions and account for any movement in the protrusions. Intensities were normalized to their initial activation intensity and analyzed in Prism. As a control, WT HUVEC, were first treated with 20 μM Y-27632 for 20 min, then a cocktail of 20 μM Y-27632, 8 μM Jasplakinolide, and 5 μM Latrunculin-B (JLY[46]) for 30 min to freeze actin turnover.

**Stimulated emission depletion (STED) microscopy.** HUVEC cells on continuous collagen were prepared as described above with the following modifications. Samples

were blocked in 3% BSA, 10% Goat Serum, and 0.1% Triton X-100 in PBS. The samples were imaged on a home-built, fast-scanning 2-color STED microscope previously described[64]. Briefly, this microscope uses pulsed, 530 nm and 635 nm excitation lasers and an 80 MHz, 750 nm depletion beam for both dyes sourced from a Ti:sapphire, mode-locked oscillator with fast-scanning achieved by a 7.5 kHz resonant mirror. The approximate resolution was measured to be ~60 nm FWHM using single molecules of Atto-647N. The microscope was controlled with LabVIEW software (National Instruments).

**Intensity correlation quotient analysis.** Intensity correlation quotient (ICQ) analysis was performed using a custom MATLAB script. First, masks of the protrusions and of regions outside of the protrusions of a similar area were hand-drawn using the imfreehand MATLAB function. Within each of these regions, the ICQ was calculated for each set of two-color STED images following the equation defined previously[52], and a signal threshold of 1 photon with positive ICQ values indicating colocalization. To determine the ICQ value of the randomly distributed case, the F-actin images were pixel-wise scrambled within the masked regions and the ICQ was calculated between these scrambled F-actin images and the original T-Plastin images.

**Platinum replica electron microscopy (PREM), and protrusion depth analysis.** WT and T-plastin KO HUVEC cells were plated on coverslips coated with 30 μg/mL collagen and cultured overnight. Cells were extracted with 1% Triton X-100 in PEM buffer (100 mM Pipes-KOH, pH 6.9, 1 mM MgCl2, and 1 mM EGTA) containing 2% polyethelene glycol (PEG) (MW 35,000), 2 μM phalloidin and 10 μM taxol for 3 min at room temperature. After three quick rinses with PEM buffer containing 2 μM phalloidin and 10 μM taxol, the extracted cells were fixed with 2% glutaraldehyde in 0.1 M sodium cacodylate buffer (pH 7.3) for 20 min. Sample processing for PREM was performed as described previously[49,50]. Briefly, glutaraldehyde-fixed cells were post-fixed sequentially with 0.1% tannic acid and 0.2% uranyl acetate in water, dehydrated in a graded ethanol series (10%, 20%, 40%, 60%, 80%, and twice 100% ethanol for 5 min each), treated with 0.2% uranyl acetate in 100% ethanol for 20 min, and washed with 100% ethanol four times for 5 min in each. Samples were then critical-point dried, and coated with platinum and carbon. The PREM samples were examined using a JEM 1011 transmission electron microscope (JEOL USA, Peabody, MA) operated at 100 kV. Images were acquired by an ORIUS 832.10 W CCD camera (Gatan, Pleasanton, CA) and presented in inverted contrast.

The length of the branched actin network from the leading edge of protrusions was measured manually in Fiji. Briefly, fields of view were selected where the branched actin network was easily distinguished. Multiple line measurements were made at several locations in each protrusion from the leading edge to the back area where branched actin ends to account for variations within each protrusion. The measurements were plotted and statistical comparisons were performed in Prism (GraphPad)

**3D collagen invasion assay.** Invasion of HUVEC into 3D collagen gels has been elegantly described elsewhere[43]. Briefly, thin gels of 2.4 mg/mL bovine collagen I containing 1 μM sphingosine-1-phosphate (Avanti Polar Lipids, 860492P) were formed in angiogenesis slides (Ibidi, 81506). WT and KO HUVEC cells were plated on top of the gel in EGM2 supplemented with 40 ng/mL bFGF and human recombinant VEGF (Thermo Fisher Scientific, 293VE050). After 24 h, wells were fixed in 4% paraformaldehyde and stained with Alexa488-phalloidin. Confocal z-stacks were acquired and the number of invaded cells into the collagen gel were quantified manually in Fiji. Statistical comparisons were performed in Prism.

**Wound healing assay and analysis.** Confluent HUVEC plated on a collagen coated 96-well glass bottom plate were stained with the nuclear marker Hoechst, and scratched with a custom built Delrin-tipped spring-loaded scratching apparatus that generates uniform horizontal scratches across the center of each well[42]. Imaging was performed for ~16 h with acquisitions every 10 min. Segmentation of cell nuclei and tracking were performed using custom MATLAB scripts (Mathworks) and have been described previously[32]. Briefly, individual cell nuclei were identified using a modified Otsu's method[65], and between each frame nuclei were linked on the basis of the nearest-neighbour method. If mitosis occurred, one of the daughter cells was directly linked to the mother cell. The other was assigned a new identifying number, and its correlation with the mother cell was saved in a separate file. Single-cell velocities were determined on the basis of nuclear displacements between subsequent time points and averaged. Plotting and statistics were performed in Prism.

**Ratiometric image analysis and cross-correlation.** Ratiometric image analysis of the RaichuCdc42 sensor and cells expressing T-Plastin, myosin light chain (MYL9), and F-actin reporters was performed with custom MATLAB scripts which have been previously described[22]. Briefly, if channels were acquired simultaneously using two cameras they were aligned via coordinate-mapping to achieve perfect registration[22]. Cell area masks were identified and segmented using automated intensity thresholding, and images were subjected to local background subtraction.

Then, ratiometric values for FRET or T-Plastin/MYL9 intensities were computed by dividing by CFP or F-tractin intensities, respectively.

Cell edge dynamics were calculated using MATLAB routines described previously[22]. Briefly, the boundary of the cell of 10 pixels in depth was divided into windows. For each window, the T-Plastin/F-tractin ratio was computed as the mean of all the pixels within that window. The local protrusion/retraction velocity was determined by tracking boundary points between frames by minimizing the squared sum of distances between matched points while preserving the order of points along the cell periphery. Cross-correlation analysis of T-Plastin, F-tractin, and cell protrusion was a slightly modified version of a previously described analysis script[22]. Briefly, signals were vectorized and then Pearson's correlation between the corresponding intensity and cell protrusion was calculated for individual time offsets. Signals were padded with NaN values such that there was not significant correlation decay at the periphery of signals.

**Leading edge protrusion and retraction rate analysis**. HUVEC stably expressing F-tractin-mCitrine were transfected with non-targeting control siRNA or T-Plastin siRNA-2 (siT-Plastin #1) or siRNA3 (siT-Plastin #2) (Fig. S4a). After 48hrs, cells were imaged every 10 s for ~33 min. Cell masks for each frame were generated in MATLAB and a time encoded projection of every 5th frame was made using a parula colormap in FIJI. Kymographs were generated using the KymographBuilder feature in FIJI. Protrusions and Retractions were traced with free-hand lines in FIJI and the X–Y coordinates were fit to a line in Excel. The slopes of the lines, representing rates of protrusion or retraction, were plotted and analyzed in Prism.

**F-actin and myosin line intensity analysis**. To compare levels of myosin depletion at the leading edge of HUVEC treated with either control or T-Plastin-targeting siRNA cells expressing stoichiometric levels of MYL9-mTurquoise and F-tractin-mCherry and treated with siRNA were imaged every 10 s to ensure edges used for quantification were located at the migratory front of cells. Intensities of MYL9 and F-tractin were quantified from a line drawn from the very front of the cell edge toward the stress fibers located in the lamellum region for ~10 migrating cells. Normalized median intensity values were calculated for each position and fit to a LOWESS-spline fitting function. The cell edge was defined as the first peak of F-tractin from the front, the region of myosin activity was defined as the beginning inflection point of an increased MYL9 slope in the fit.

**Protrusion width analysis**. WT and T-Plastin KO HUVEC were sparsely plated, then fixed and stained for F-actin and Arp3. Protrusions were defined by their Arp3 intensity and regions were manually drawn across the width of Arp3-based protrusions using the ROI manager in FIJI. The measured widths of these protrusions were then analyzed in Prism.

**Focal adhesion kinetic analysis**. WT and T-Plastin KO HUVEC stably expressing F-tractin-mCitrine were transfected with mCherry-Paxillin and imaged after 24 h. Frames were acquired every 5 s for ~25 min and the assembly and disassembly rates of focal adhesions were quantified using an online analysis tool described previously[47,48]. Rates were then plotted and analyzed using Prism.

**Fibronectin micropattern generation**. Pre-labeled rhodamine fibronectin was from Cytoskeleton Inc (FNR01). To generate Alexa647-labeled fibronectin, bovine fibronectin from Sigma (F4759) was labeled using Alexa647-NHS Ester (Thermo Fisher Scientific) according to the manufacturer's instructions and run over a PD-10 column (GE Lifesciences) to remove unconjugated dye.

96-well glass bottom plates (Cellvis) were plasma cleaned under vacuum (Harrick Plasma) and incubated in 0.1 mg/ml PLL-PEG (SuSoS) in 10 mM HEPES buffer for at least 1 h. After washing with PBS and addition of pLPP (NanoscaleLabs), pattern regions were locally exposed to UV light using the Primo system (Alveole) on an automated Leica DMi8 microscope with 20 × 0.4NA objective and equipped with an ORCA flash 4.0 sCMOS camera (Hamamatsu). Wells were then washed with PBS again and incubated for 30 min with either rhodamine or Alexa647-labeled fibronectin diluted into unlabeled fibronectin at a final concentration of 50 μg/mL. Wells were then washed in PBS, sealed with adhesive foil, and stored at 4 °C until use.

**Micropattern imaging and analysis**. HUVEC stably expressing F-tractin-mCitrine were added to patterns of labeled fibronectin and a total area consisting of four patterns was imaged across 12–15 camera field of views every 30 s. The field of views were registered and combined as a montage using Slidebook software (Intelligent Imaging Innovations) and exported for analysis with a custom MATLAB script. Cells on stripes of fibronectin were detected and segmented via automatic thresholding. The change in cell area was calculated after 1 h of a cell making full contact with a region of fibronectin, and these values were automatically grouped together based on what fibronectin gap spacing lane they made contact with. If cells made contact with each other, or if another cell occupied a neighboring fibronectin stripe during this one-hour period after first contacting

fibronectin, the cell was excluded from the area change analysis. To analyze the dynamics of cell area changes, the area of a cell relative to its initial area upon contact with a fibronectin stripe was measured over time.

Quantification of the change in number of stripes occupied was performed manually in Slidebook. Briefly, Cells were manually curated to ensure they had contact with fibronectin stripes at least 30 min prior to treatment and did not come into contact with another cell. After treatment with either imaging buffer (control), CK666, Blebbistatin and Y-27632, or deionized $H_2O$, the initial number of fibronectin stripes a cell occupied prior to treatment was subtracted from the number of stripes it made contact with 45 min after treatment. These values were then plotted and analyzed in Prism.

**Phylogenic analysis**. Protein sequences for a subset of Plastin orthologues were aligned using the MUSCLE method in MegAlign Pro (DNASTAR Inc.) and exported for visualization using the interactive tree of life (iTOL, https://itol.embl.de).

**Statistics and reproducibility**. Differences in cell area change on micropatterns were analyzed using either a one-way ANOVA with either Dunnett's or Sidak's multiple comparison test for drug treatments or with multiple $t$ tests for comparing the different gap sizes in Prism (Graphpad). Differences in cell migration rates of scratch assays were tested with a one-way ANOVA with Sidak's multiple comparison test in Prism. Statistically significant differences are indicated as follows: *$P < 0.05$, **$P < 0.01$, ***$P < 0.001$, ****$P < 0.0001$. All results were representative of at least two independent experiments.

**Reporting summary**. Further information on research design is available in the Nature Research Reporting Summary linked to this article.

## Data availability
All data that support the conclusions are available from authors upon reasonable request. A reporting summary for this article is available as a Supplementary Information file. Source data are provided with this paper.

## Code availability
All MATLAB scripts used in this study are available at: https://github.com/dgarbett/Garbett-et-al-2020-Code.

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

## Acknowledgements

We are grateful to all members of the Meyer, Moerner, and Svitkina labs for helpful comments and discussions. We thank Yilin Fan for help with code optimization for edge detection for the scratch assay analysis. We are also grateful to Nalin Ratnayeke for statistics advice. We also thank Dr. Daniel Cohen and Dr. Gaspard Pardon for help and suggestions with the Alvéole Primo system, and Jaime Larios for technical support. We are also grateful to Dr. Mingyu Chung, Dr. Lindsey Pack, and Dr. Marielle Köberlin for critically reading this paper. Funding provided by GM127026 (T.M.). D.G. was funded in part by GM116328. Additional funding provided by R01-GM 095977 (T.M.S.), and R35-GM118067 (W.E.M.). D.G.M. was supported by NSF-GRFP. The project described was supported, in part, by Award Number 1S10OD01227601 from the National Center for Research Resources (NCRR). Its contents are solely the responsibility of the authors and do not necessarily represent the official views of the NCRR or the National Institutes of Health.

## Author contributions

D.G. and T.M. conceptualized the study and methodology, acquired funding, and wrote the paper. D.G., C.Y., and D.G.M. performed the investigation, validation, visualization, and data curation. D.G., A.B., and D.G.M. performed the formal analysis of the data. D.G., A.B., A.H., and D.G.M. developed software for the analysis. D.G. and A.H. developed resources for the study. D.G., A.B., A.H., C.Y., D.G.M., T.M.S., W.E.M., and T.M. reviewed and edited the paper. T.M. supervised the study and administrated the project.

## Competing interests

The authors declare no competing interests.

**Additional information**

