## [Peer Review File · Nature Communications]

REVIEWER COMMENTS

Reviewer #2 (Remarks to the Author):

This work deals with the role of T-plastin in stabilizing cell protrusions so that cells are able to bridge gaps in the ECM and continue forward motion. It is a re-submission. The authors have done a good job in responding to the previous criticisms. I also note that the manuscript has transferred from NCB to Nat Comms. I am now highly supportive of publication in Nat Comms as the work is performed to a high standard and has interesting new information. I would suggest only minor changes.

Specific comments

- 1) The figure legends are overly brief and should be expanded.
- 2) The data in SF 4C is presented in a confusing manner. I can well believe that scratch assays perturb the substrate, but I am confused as to whether the authors are arguing that there is less matrix (collagen) or that it is differently organised. Why is there no collagen in the non-scratched area that is not covered by cells? Images of assays that have never been scratched would be helpful for comparison - or is the argument that these will be like the areas with cells? Either way, clarification is needed.

Reviewer #3 (Remarks to the Author):

Review of Garbett et al.

The authors have tried to address some of the major points brought up in the previous version of the manuscript. However, the additional experiments and the revised manuscript still do not address the major gap in this study about the mechanism by which T-Plastin regulates matrix gap-bridging during cell migration. My biggest concerns are related to the emphasis made on the role of bundling of T-plastin in bridging ECM gaps. To me, it is clear that there is a leading-edge phenotype when T-Plastin is lost, but it is not clear if the phenotype observed are due to loss of actin bundles or through changes in other lamellipodial interactions or coupling of actin with adhesions. One excellent addition is the ultrastructure EM data in the new version which clearly shows that loss of T-Plastin results in reduction the lamellipodia width, but again, how reduction in lamellipodia width is related to loss of bundling is unclear. This and other issues are described below:

Major points:

1. Immunostaining of T-plastin on cells on ECM ridge- In figure 2B, the authors show beautiful staining of the endogenous T-Plastin at the edge of a wound. The authors also show beautiful IF staining of actin, fibronectin and paxillin on ECM ridges (Fig 1G). However, unless I am missing something, it would be critical to see T-Plastin IF on ECM ridges. In fig 4A, they show how expressed

protein looks like on ridges and ofcourse the background is high on the images and the resolution low to see details. Is there a technical reason for not having IF staining with Ab for endogenous protein on ECM ridges? If not, it needs to be included.

2. Role of T-plastin in protrusions- Is it bundling defect? In figure 2, the authors nicely show T-Plastin in the lamellipodia and coincident with active protrusions. They also show changes in Arp2/3 localisation and actin staining when you knock T-Plastin down. I think it's critical to show changes in leading edge dynamics on the loss of T-Plastin and characterise it. Others, including papers from Sheetz, Bear, Haugh etc have shown that the lamellipodia can buckle (or not adhere depending on the defect) if not stabilised mechanically. In the case of T-Plastin here, we don't know if lamellipodia is being formed with normal dynamics and then not stabilised (which would suggest a role for T-plastin in stabilisation) or if the lamellipodia is not being formed to begin with, which is a different phenotype. As stated above, I am still unsure about the conclusion made by the authors throughout the text and specifically from Fig 4A. All observation could still be due to loss of lamellipodia phenotype which could be because of a variety of reasons and functions of T-Plastin (including potentially bundling!). The EM images don't necessarily reveal that either, so this would require a lot of toning down throughout the text or some direct data showing loss of actin bundles or other experiments specifically related to bundling. If the conclusion of bundling is because of no difference in polymerisation rate measured in Figure 5A and B, then that is still weak and definitely should not be brought up in the text till describing the results of Figure 5. Would still require in my opinion toning down of the bundling part throughout the text.

3. I think it would be good to have a movie with the Plastin KO on the ECM ridges of different gaps to really appreciate the differences in the bridging phenotype since the quantification through the change in area parameter is subtle.

4. Adhesion maturation defect: The role of adhesions in this process is still not clear. NA and FA being closer to each other can be due to loss of lamellipodia thickness as the authors suggest which could be either due to faster maturation to nascent adhesions to focal adhesions or due to the lamellipodia defect. In the blebbistatin phenotype, in WT cells, does T-plastin and the lamellipodia become broader? As in point 2, imaging of the leading edge with an adhesion marker can immediately show the differences in adhesion and leading-edge dynamics due to loss of T-Plastin does clearing any doubts about contribution of adhesion maturation (a myosin II dependent process) or an T-Plastin lamellipodia process

Minor points:

1. In fig 4A, the gap size being used is not mentioned at all. I am guessing its 4um gap size.
2. Role of membrane tension: In figure 4D, the authors show the rate of increase of area of cells on 4um and 8um gaps and show that the initial rate is quite similar between WT and KD cells. On 8um gaps, that difference still seems quite similar which is also ok and consistent with the smaller difference seen in Figure 4C. However, the authors state that the later divergence observed is due to potentially membrane tension forces becoming limiting. Is that consistent with rate of area change measured with WT cells treated with MBCD?

Reviewer #4 (Remarks to the Author):

Technical review for "T-Plastin reinforces membrane protrusions to bridge matrix gaps during cell migration".

The work described in the manuscript by the Meyer group in collaboration with the Svitkina group proposes that a protein called T-plastin has a crucial role for cell migration accross gaps. The authors propose that the actin bundling protein T-Plastin widens and lengthens protrusions and is specifically enriched in active protrusions where F-actin is devoid of non-muscle myosin II activity. More specifically concerning the electron microscopy analysis, the authors used metal replica EM to show that contrarily to the thick branched actin network present at the leading edge of WT cells, this network is much thinner in T-Plastin knock-out cells. In addition, they show that typical straight filopodia emanating from lamellipodial protrusions in WT cells, are more bent in the T-Plastin KO cells. The authors overall conclusion suggest that the protrusive actin networks of filopodia and lamellipodia are less resilient to forces of membrane tension in the absence of T-Plastin's bundling activity.

major comments:

The Svitkina laboratory is one of the very few recognized experts capable of visualizing actin networks in lamellipodia and filopodia at the ultrastructural level by metal replica electron microscopy. The results proposed in this manuscript are original and of high quality; they metal replica experiments are convincing and have been quantified to support the visual claims.

My main comment concerning the electron microscopy part would be to include both lower and higher magnification images in the main figures. It is quite difficult for readers who are not familiar with the technique to understand where the high(medium) magnification images come from and quite a shame to have to search the supplement to find the ultrastructural context of the images in Fig. 6. Higher magnification images focusing on the actin network should also be shown to build-up on further (already available) explanations on whether it is only the size of the protrusion depth that is decreased or if and how T-Plastin absence modifies the molecular organization of these networks. The absence or presence of ultrastructural differences between the branched actin networks at these protrusions could be discussed further.

minor comments:

-On p.9 line 32 it is stated that "but in KO cells, these filopodia were more often bent (Fig. SBC, arrows)" which should read (Fig. 6BC, arrows).

Response to reviewers comments:

Reviewer 1 (via Reviewer 2):

“...suggested statistical analyses be explicit in the figure/figure legend of Figure 4G.”

We have now added statistical information to this subpanel and legend, we apologize for this oversight. Please note this panel is now located in Fig 5F.

Reviewer 2:

“...The authors have done a good job in responding to the previous criticisms. I also note that the manuscript has transferred from NCB to Nat Comms. I am now highly supportive of publication in Nat Comms as the work is performed to a high standard and has interesting new information. I would suggest only minor changes.”

We thank this reviewer again for their helpful comments and enthusiasm regarding this study. We have made the suggested minor changes as detailed below

“1) The figure legends are overly brief and should be expanded.”

We have now expanded our figure legends to be more descriptive and easier to follow while also being mindful of space limitations.

“2) The data in SF 4C is presented in a confusing manner. I can well believe that scratch assays perturb the substrate, but I am confused as to whether the authors are arguing that there is less matrix (collagen) or that it is differently organised. Why is there no collagen in the non-scratched area that is not covered by cells? Images of assays that have never been scratched would be helpful for comparison - or is the argument that these will be like the areas with cells? Either way, clarification is needed.”

This is a good point. We have rearranged the figure, and revised the text and figure legend to make the conclusions more clear and have added an additional panel showing collagen distribution alone in the absence of cells (Fig. S6A-C). Although it might be difficult to see in the compressed figure files, there is still dim collagen visible in the non-scratched area that is not covered by cells (collagen staining inside cells is much brighter). Our point here was merely to illustrate that these type of scratch assays can disrupt ECM, a fact not necessarily understood by groups performing these assays. These changes should clarify this point.

Reviewer 3:

“...the additional experiments and the revised manuscript still do not address the major gap in this study about the mechanism by which T-Plastin regulates matrix gap-bridging during cell migration. My biggest concerns are related to the emphasis made on the role of bundling of T-plastin in bridging ECM gaps. To me, it is clear that there is a leading-edge phenotype when T-Plastin is lost, but it is not clear if the phenotype observed are due to loss of actin bundles or through changes in other lamellipodial interactions or coupling of actin with adhesions.”

We appreciate this reviewer's concern regarding the role of T-Plastin actin bundling in the ECM gap bridging process. As detailed below, we have altered the text to make it clear we have not tested actin bundling directly in our experiments. Instead we discuss that T-Plastin and other plastins have been extensively studied in biochemical and cellular assays in different model organisms and the only well-established role of plastins is their activity in bundling actin filaments. Our study is consistent with such a bundling role of T-Plastin and we now make it more clear that we do not directly demonstrate bundling. However, we now more extensively discuss throughout the manuscript how our cellular findings can plausibly be interpreted by the known bundling activity role of T-Plastin without excluding potential additional functions. Further, we have now added a substantial amount of new figure panels addressing in more detail changes in focal adhesion and leading edge dynamics in T-Plastin deficient cells, and other concerns as described in the specific points below.

"One excellent addition is the ultrastructure EM data in the new version which clearly shows that loss of T-Plastin results in reduction the lamellipodia width, but again, how reduction in lamellipodia width is related to loss of bundling is unclear. This and other issues are described below:"

We are grateful for this reviewer's enthusiasm regarding the ultrastructural data and their previous suggestion of including this type of analysis, which has greatly improved our manuscript. We have added additional panels (Fig. 7A) further demonstrating the reduction of F-actin at the leading edge seen by ultrastructural EM as per reviewer 4's suggestion.

"Major points:

1. Immunostaining of T-plastin on cells on ECM ridge- In figure 2B, the authors show beautiful staining of the endogenous T-Plastin at the edge of a wound. The authors also show beautiful IF staining of actin, fibronectin and paxillin on ECM ridges (Fig 1G). However, unless I am missing something, it would be critical to see T-Plastin IF on ECM ridges. In fig 4A, they show how expressed protein looks like on ridges and of course the background is high on the images and the resolution low to see details. Is there a technical reason for not having IF staining with Ab for endogenous protein on ECM ridges? If not, it needs to be included."

This is an excellent point. We have now included images showing the endogenous localization of T-Plastin in HUVEC on the ECM ladder patterns (Fig. S4A and B). As the reviewer pointed out, there is a higher background seen with T-Plastin staining in cells on the ECM ladder patterns, however we still see clear enrichment in protrusions that bridge ECM gaps (marked by asterisks).

"2. Role of T-plastin in protrusions- Is it bundling defect? In figure 2, the authors nicely show T-Plastin in the lamellipodia and coincident with active protrusions. They also show changes in Arp2/3 localisation and actin staining when you knock T-Plastin down. I think it's critical to show changes in leading edge dynamics on the loss of T-Plastin and characterise it. Others, including papers from Sheetz, Bear, Haugh etc have shown that the lamellipodia can buckle (or not adhere depending on the defect) if not stabilised mechanically. In the case of T-Plastin here, we don't know if lamellipodia is being formed with normal dynamics and then not stabilised (which would suggest a role for T-plastin in stabilisation) or if the lamellipodia is not being formed to begin with, which is a different phenotype. As stated above, I am still unsure about the conclusion made by the authors throughout the text and specifically from Fig 4A. All observation could still be due to loss of lamellipodia phenotype which could be because of a variety of reasons and functions of T-Plastin (including potentially bundling!). The EM images don't necessarily reveal that either, so this would require a lot of toning down throughout the text or some direct data

showing loss of actin bundles or other experiments specifically related to bundling. If the conclusion of bundling is because of no difference in polymerisation rate measured in Figure 5A and B, then that is still weak and definitely should not be brought up in the text till describing the results of Figure 5. Would still require in my opinion toning down of the bundling part throughout the text.”

This reviewer makes another good point here. We have now added an additional new figure (Fig. 3) examining how lamellipodial dynamics change in the absence of T-Plastin. We quantified the protrusion and retraction rates across many cells in a similar manner to that used in studies by the Bear, Haugh, and Sheetz labs. We find that loss of T-Plastin reduces both protrusion and retraction rates, suggesting that leading edge dynamics are overall reduced in the absence of T-Plastin. Additionally, we have added example images and supplemental movies highlighting the behavior of T-Plastin KO cells on ECM ladder patterns (Fig. S4F, Movies S6 and S7). We see that T-Plastin KO cells show similar protrusive behavior to WT cells in that they still form lamellipodia and filopodia, however they fail to increase their cell surface area efficiently compared to WT cells while spreading across ECM gaps.

The reviewer is also correct that we haven't directly tested if T-Plastin bundling activity itself is responsible for its role in protrusions or ECM gap bridging. Previous biochemical studies have well characterized T-Plastin as an actin bundler, but directly showing loss of bundles (and not just actin filaments) in cells has proved quite difficult for us and others. The ultrastructural EM images clearly show a reduction in the depth and density of the actin network at the leading edge, however we cannot definitively observe bundled vs unbundled in the cellular experiments. As this reviewer suggests, we have altered the text to more accurately describe that we are not directly testing bundling, but instead performing cellular assays involving T-Plastin and placing those findings in the context of T-Plastin's well established role as an actin bundling protein.

“3. I think it would be good to have a movie with the Plastin KO on the ECM ridges of different gaps to really appreciate the differences in the bridging phenotype since the quantification through the change in area parameter is subtle.”

This is a great idea, and we should have included this before. We have now added additional examples of the ECM Gap bridging process (Fig. S4F, Movies S6 and S7). The images and movies show that T-Plastin KO cells are still able to form protrusions (both filopodia and lamellipodia) but that they struggle to achieve the same change in cell spreading area compared to WT cells, suggesting their protrusive behavior is less efficient in this process.

“4. Adhesion maturation defect: The role of adhesions in this process is still not clear. NA and FA being closer to each other can be due to loss of lamellipodia thickness as the authors suggest which could be either due to faster maturation to nascent adhesions to focal adhesions or due to the lamellipodia defect. In the blebbistatin phenotype, in WT cells, does T-plastin and the lamellipodia become broader? As in point 2, imaging of the leading edge with an adhesion marker can immediately show the differences in adhesion and leading-edge dynamics due to loss of T-Plastin does clearing any doubts about contribution of adhesion maturation (a myosin II dependent process) or an T-Plastin lamellipodia process.”

The reviewer makes a good point regarding the potential for altered focal adhesion dynamics in the absence of T-Plastin. We have now quantified the assembly and disassembly rates of focal adhesions in protrusions using the identical online analysis tool previously used by James Bear's group (Wu et al 2012, Berginski et al 2011). The new figure (Fig. S7) shows no statistically significant difference in either the assembly or disassembly rates of focal adhesions in T-Plastin KO cells compared to WT. This, combined

with the new edge dynamic analysis shown in Fig. 3, shows that loss of T-Plastin significantly reduces leading edge dynamics without having a significant effect on adhesion assembly or disassembly. We are grateful for this reviewer's line of thinking regarding adhesions and protrusion dynamics, and feel these valuable additions have substantially improved the manuscript.

We have also added an additional figure panel demonstrating the effects of blebbistatin treatment on T-Plastin and lamellipodia (Fig. S3B). We would like to note that there is no immediate effect of blebbistatin (shown at 50 s) on leading edge F-actin and T-Plastin staining. We think the delayed effect on increased protrusion and T-Plastin visible in the 8-15 minute images is caused by indirect effects whereby cells have less actomyosin contraction elsewhere in the cell.

“Minorpoints:

1. In fig 4A, the gap size being used is not mentioned at all. I am guessing its 4um gap size.”

These are indeed 4 um gaps, we apologize for this oversight and have now added this information to the figure legend (as seen in new Fig. 5A).

“2. Role of membrane tension: In figure 4D, the authors show the rate of increase of area of cells on 4um and 8um gaps and show that the initial rate is quite similar between WT and KD cells. On 8um gaps, that difference still seems quite similar which is also ok and consistent with the smaller difference seen in Figure 4C. However, the authors state that the later divergence observed is due to potentially membrane tension forces becoming limiting. Is that consistent with rate of area change measured with WT cells treated with MbCD? “

This is a great point. We have now added an additional panel showing the altered kinetics of MbCD treatment in WT and T-Plastin KO cells (Fig. S4H). MbCD treatment removes the burst in spreading typically seen between 5-20 minutes after a cell first makes contact with the ECM ladder. The effects of MbCD on the plasma membrane can potentially be interpreted in different ways, so we were careful to not over-interpret these results. That being said, the lack of a protrusion burst after MbCD addition suggests that membrane protrusion is significantly reduced due to increased tension created by MbCD, arguing that membrane tension does play a limiting role - and such a role should be particularly relevant at later stages when larger cell surface areas create increased tension.

Reviewer 4:

“The Svitkina laboratory is one of the very few recognized experts capable of visualizing actin networks in lamellipodia and filopodia at the ultrastructural level by metal replica electron microscopy. The results proposed in this manuscript are original and of high quality; they metal replica experiments are convincing and have been quantified to support the visual claims.”

We thank this reviewer for their enthusiasm!

“My main comment concerning the electron microscopy part would be to include both lower and higher magnification images in the main figures. It is quite difficult for readers who are not familiar with the technique to understand where the high(medium) magnification images come from and quite a shame to have to search the supplement to find the ultrastructural context of the images in Fig. 6. Higher

magnification images focusing on the actin network should also be shown to build-up on further (already available) explanations on whether it is only the size of the protrusion depth that is decreased or if and how T-Plastin absence modifies the molecular organization of these networks. The absence or presence of ultrastructural differences between the branched actin networks at these protrusions could be discussed further.”

The reviewer makes an excellent point. We have now included the lower magnification images depicting where in a cell the higher resolution leading edge images are located to better orient readers unfamiliar with these types of images (Fig. 7A). Additionally, we have added higher magnification images of regions within 1 μm of the leading edge to give a clearer depiction on the architecture of the actin networks in WT and KO cells. We also have added to the text to better describe the reduced density of F-actin seen in these regions.

“minor comments:

-On p.9 line 32 it is stated that "but in KO cells, these filopodia were more often bent (Fig. SBC, arrows)" which should read (Fig. 6BC, arrows).”

We apologize for this oversight and have fixed this in the revised manuscript. Please note this panel is now located as Fig. S8A.

REVIEWERS' COMMENTS

Reviewer #3 (Remarks to the Author):

The authors have addressed all my queries and I thank them for diligently doing so. It is a beautiful piece of work and really adds to the intrigue of the mechanical role of the lamellipodia during cell migration. Someone I trained under always said that the lamellipodia is a mechanosensory organelle (this was in context of coupling between nascent adhesions and the lamellipodia actin) the potential implication of this study including how actin regulation itself could be sensitive to ECM cues is important. Thank you for the work and stay safe.

Reviewer #4 (Remarks to the Author):

The authors have satisfactorily addressed my comments.

Reviewer #3 (Remarks to the Author):

“The authors have addressed all my queries and I thank them for diligently doing so. It is a beautiful piece of work and really adds to the intrigue of the mechanical role of the lamellipodia during cell migration. Someone I trained under always said that the lamellipodia is a mechanosensory organelle (this was in context of coupling between nascent adhesions and the lamellipodia actin) the potential implication of this study including how actin regulation itself could be sensitive to ECM cues is important. Thank you for the work and stay safe.”

We thank this reviewer again for their enthusiasm and helpful comments which have greatly improved the manuscript

Reviewer #4 (Remarks to the Author):

The authors have satisfactorily addressed my comments.

We appreciate this reviewer’s helpful comments in regards to best presenting our PREM images.